# Parabrachial opioidergic projections to preoptic hypothalamus mediate behavioral and physiological thermal defenses

Aaron J Norris[1†]*, Jordan R Shaker[2†], Aaron L Cone[1], Imeh B Ndiokho[1], Michael R Bruchas[3]*

[1]Department of Anesthesiology, Washington University School of Medicine, St. Louis, United States; [2]Medical Scientist Training Program, University of Washington, Seattle, United States; [3]Center for the Neurobiology of Addiction, Pain and Emotion, Departments of Anesthesiology and Pharmacology, University of Washington, Seattle, United States

*For correspondence:
norrisa@wustl.edu (AJN);
mbruchas@uw.edu (MRB)

[†]These authors contributed equally to this work

Competing interests: The authors declare that no competing interests exist.

**Abstract** Maintaining stable body temperature through environmental thermal stressors requires detection of temperature changes, relay of information, and coordination of physiological and behavioral responses. Studies have implicated areas in the preoptic area of the hypothalamus (POA) and the parabrachial nucleus (PBN) as nodes in the thermosensory neural circuitry and indicate that the opioid system within the POA is vital in regulating body temperature. In the present study we identify neurons projecting to the POA from PBN expressing the opioid peptides dynorphin and enkephalin. Using mouse models, we determine that warm-activated PBN neuronal populations overlap with both prodynorphin (Pdyn) and proenkephalin (Penk) expressing PBN populations. Here we report that in the PBN *Prodynorphin* (*Pdyn*) and *Proenkephalin* (*Penk*) mRNA expressing neurons are partially overlapping subsets of a glutamatergic population expressing *Solute carrier family 17* (*Slc17a6*) (VGLUT2). Using optogenetic approaches we selectively activate projections in the POA from PBN Pdyn, Penk, and VGLUT2 expressing neurons. Our findings demonstrate that Pdyn, Penk, and VGLUT2 expressing PBN neurons are critical for physiological and behavioral heat defense.

## Introduction

Maintaining body temperature in the face of changing environmental conditions is a core attribute of mammals, including humans, and is critical for life. Achieving a stable body temperature requires information about the temperature of the periphery and environment to be integrated to drive physiological and behavioral programs to defend the core temperature (*Jessen, 1985*). Physiological parameters modulated to maintain temperature include thermogenesis (utilization of brown adipose tissue [BAT], shivering), changes in circulation (vasodilation and vasoconstriction), and evaporation (*Cabanac, 1975*). Behavioral modifications include selection, when possible, of ambient temperature, altering posture to alter heat loss, and modulation of physical activity level. Responding to thermal challenges involves perception of temperature, encoding the valence of the temperature (e.g. too hot), and evoking appropriate physiological responses (*Tan and Knight, 2018*). Perceptive, affective, and autoregulatory elements may be encoded by overlapping or discrete neuronal circuits. The preoptic area of the hypothalamus (POA) and the parabrachial nucleus (PBN) have been identified as key nodes within the neurocircuitry regulating body temperature. In the report presented

here, we identify and delineate the unique roles of genetically defined neuronal populations in PBN projecting to the POA in responding to environmental warmth.

The POA contains neurons critical for integration of information about body temperature and for coordination of responses to thermal challenges to maintain core temperature (*Abbott and Saper, 2017*; *Abbott and Saper, 2018*; *Tan et al., 2016*). Neurons in POA, identified by different genetic markers, can regulate BAT activation, drive vasodilation, and shift ambient temperature preferences (*Tan et al., 2016*; *Yu et al., 2016*). Prior evidence has suggested critical roles for inputs from the PBN to the POA in regulating temperature (*Geerling et al., 2016*; *Miyaoka et al., 1998*; *Morrison, 2016*). The PBN is, however, a highly heterogenous structure with subpopulations known to relay various sensory information from the periphery (thirst, salt-appetite, taste, pain, itch, temperature, etc.) and playing key roles in nocifensive responses, specifically escape and aversive learning (*Chiang et al., 2020*; *Kim et al., 2020*; *Palmiter, 2018*). The studies here examine the roles for parabrachial glutamatergic neurons expressing the opioid peptides dynorphin and enkephalin.

Regulation of body temperature requires integration of homeostatic and environmental inputs across varying time scales creating opportunities for neuromodulatory signaling to play key roles. In vivo experiments suggest that opioid neuropeptides, as a neuromodulator, may play a critical role in thermal homeostasis. Pharmacologic manipulation of opioid systems induces changes in body temperature and can impair thermoregulatory control in humans, rats, and other animals (*Chen et al., 2005*; *Ikeda et al., 1997*; *Spencer et al., 1990*). Opioid receptor signaling within the POA has been implicated in modulating body temperature, but potential sources for native ligands remain to be identified (*Baker and Meert, 2002*; *Clark, 1979*). Activation of mu receptors in the POA can drive opposing effects on body temperature. A recent study indicated that neurons in PBN expressing prodynophin (Pdyn), which is processed to dynorphin, the endogenous ligand for the kappa opiate receptor (KOR), are activated by ambient warmth (*Chavkin et al., 1982*; *Geerling et al., 2016*). PBN neurons expressing the endogenous mu and delta opioid receptor ligand, enkephalin, have not been examined in relation to how they may regulate temperature.

In this study we used a series of modern anterograde and retrograde viral approaches to determine the connection of PBN neurons expressing Pdyn (Pdyn+) and Penk (Penk+), to the POA (*Henry et al., 2017*). We delineate the overlap of the neuronal populations expressing these peptides with warm-activated PBN neurons. We identify subsets of Pdyn+ and Penk+ neurons that project to POA from the PBN. We then combine optogenetic and chemogenetic tools with Cre driver mouse lines to determine the causal roles of PBN neurons that project to the POA in mediating physiological and behavioral responses to thermal challenge. Here we also examine potential roles of opioid receptor mediated behaviors in both Pdyn+ and Penk+ PBN-POA projections. We report that glutamatergic, Pdyn+, and Penk+ neuronal populations projecting from PBN to POA initiate physiological and behavioral heat defensive behaviors. Chemogenetic inhibition of glutamatergic PBN neurons blocks vasomotor responses to thermal heat challenge. The studies reported here provide new insights into the thermoregulatory properties of parabrachial neuropeptide-containing projections to the hypothalamus in homeostatic and metabolic behavior.

## Results

### Ambient warmth activates Pdyn+ and Penk+ neurons in PBN

Effects of mu and kappa receptor signaling on body temperature have been described and mRNA for *Pdyn* and *Penk* has been reported to be expressed in the PBN (*Baker and Meert, 2002*; *Chen et al., 2005*; *Clark, 1979*; *Engström et al., 2001*; *Hermanson and Blomqvist, 1997*; *Hermanson et al., 1998*). To examine if PBN neurons expressing dynorphin or enkephalin opioid neuropeptides are activated by ambient warmth, we exposed mice to ambient warmth (38°C) or room temperature (21–23°C) for 4 hr prior to preparation of brain for Fos staining. We performed immunohistochemistry (IHC) on collected brains sections containing the PBN with antibody directed against Fos (anti-Fos) to examine induction of Fos expression as a marker of neuronal activation (*Sheng and Greenberg, 1990*). Consistent with recent reports, we observed induction of Fos expression in the lateral PBN (LPBN) (*Figure 1B,C*; *Geerling et al., 2016*). In brain sections from warm exposed mice (n = 8) compared to room temperature controls (n = 4), Fos staining revealed a robust and significant (p=0.003) increase in mean ± SEM number of neurons positive for Fos

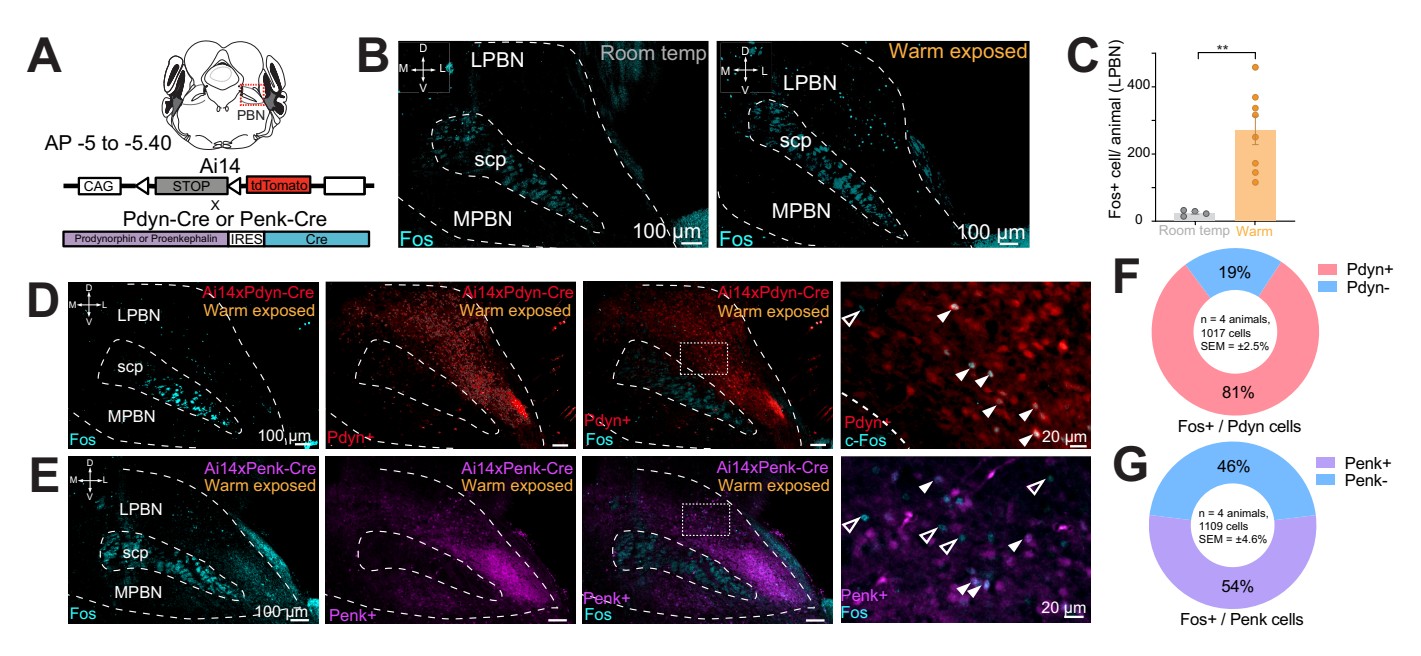

**Figure 1.** Warm-activated neurons in parabrachial nucleus (PBN) overlap with Pdyn and Penk expression. (A) Schematized view of PBN regions analyzed for Fos expressing neurons and the genetic cross schemes of Ai14xPdyn-Cre/Ai14xPenk-Cre reporter mouse lines used. (B) Representative images of brain sections harvested from animals exposed to room temperature or 38°C and probed with anti-Fos. Brains from 38°C exposed mice had significantly more neurons in PBN positive for Fos staining. (C) Quantification of Fos positive LPBN neurons per brain. Data are presented as mean ± SEM; n = 4 animals in room temp group, n = 8 animals in warm exposed group; t-test, **p<0.01. (D) Representative images of Fos labeling (cyan) in Ai14 x Pdyn-Cre brains with Fos labeling of Pdyn+ (red) (filled arrows) neurons and Pdyn- (open arrows). (E) Representative images of Fos labeling in Ai14xPenk-Cre brains with Fos labeling of Penk+ (magenta) (filled arrows) and Penk- neurons (open arrows). (F and G) Quantification of the overlap of Fos staining in Ai14xPdyn-Cre and Ai14xPenk-Cre brains demonstrated 81% or 46% of Fos cells were also overlapped with tdTomato expression in Ai14xPdyn-Cre or Ai14xPenk-Cre brains, respectively.

The online version of this article includes the following figure supplement(s) for figure 1:

**Figure supplement 1.** Validation of Penk-Cre and Pdyn-Cre lines in the parabrachial nucleus (PBN).

expression in the LPBN per brain: 265.8 ± 41.9 in warm exposed mice compared to 23.2 ± 4.0 in room temperature controls (*Figure 1C*). Cells in LPBN, lateral to superior cerebellar peduncle, in sections corresponding −5.0 to −5.4 caudal to bregma were counted. In brains from recombinase reporter mice (Ai14) crossed to Pdyn-Cre (Ai14xPdyn-Cre) or Penk-Cre (Ai14xPenk-Cre) lines, tdTomato was robustly expressed in LPBN indicating expression of Pdyn (*Al-Hasani et al., 2015*; *François et al., 2017*; *Krashes et al., 2014*; *Madisen et al., 2010*) and Penk (*François et al., 2017*) in LPBN neurons (*Figure 1D,E*). Cells expressing tdTomato in Pdyn-Cre mice (Pdyn+) and Penk-Cre mice (Penk+) were most abundant in the caudal LPBN. To determine the overlap of warm-activated neurons with Pdyn+ or Penk+ cells in LPBN, we exposed mice, Ai14xPdyn-Cre and Ai14xPenk-Cre, to a warm (38°C) ambient temperature for 4 hr prior to harvesting brains and used IHC on sections with anti-Fos. In the LPBN of Ai14xPdyn-Cre mice, we found that a mean ± SEM of 81% ± 2.5 of the cells positive for Fos staining were also positive for tdTomato expression (n = 4 animals, 1017 cells) (*Figure 1F*). In the LPBN of Ai14xPenk-Cre mice, an average ± SEM of 54% ± 4.6 (n = 4 animals, 1109 cells) of Fos-positive cells in warm-exposed mice were also positive for tdTomato (*Figure 1G*). We blindly sampled tdTomato neurons in the LPBN and then quantified the number of cells also labeled for Fos. In samples from Ai14xPdyn-Cre mice we found 22% ± 4 (n = 3 animals, 150 cells) overlap and from Ai14xPenk-Cre 18% ± 4 (n = 3 animals, 150 cells). These data indicate that warmth activated neurons in LPBN may co-express the neuropeptides dynorphin and enkephalin.

## Pdyn+ and Penk+ LPBN neurons project to the ventral medial preoptic area in the POA and are VGLUT2+

Next, to delineate possible overlapping expression of the neuropeptides, we used fluorescent in situ hybridization (FISH) with targeted probes for *Pdyn*, *Penk*, and *Slc17a6* and examined serial coronal brain sections encompassing the PBN. Based on previous studies implicating glutamate in LPBN thermosensory relay neurons (*Nakamura and Morrison, 2007*; *Nakamura and Morrison, 2010*), we hypothesized that the majority of *Pdyn* and *Penk* expressing (*Pdyn*+ and *Penk*+) LPBN neurons would also express *Slc17a6*, indicating they are glutamatergic. Consistent with expression patterns evident in the Ai14xPdyn-Cre and Ai14xPenk-Cre mice, *Pdyn* and *Penk* FISH probes labeled neurons in the LPBN (*Figure 2F and G*). *Pdyn*+ and *Penk*+ cells were most abundant in the caudal LPBN. Sections were also co-labeled with *Slc17a6* probes with *Pdyn* or *Penk* probes. The overlap of cells in LPBN labeled with each probe was quantified. A mean ± SEM of 98% ± 0.9 (n = 760 cells, n = 4 mice) of *Pdyn* labeled cells were positive for *Slc17a6* (*Figure 2A,I*). A mean ± SEM of 97% (n = 650 cells, n = 4 mice) of *Penk* labeled cells were positive for *Slc17a6* (*Figure 2B,J*). Surprisingly, a mean ± SEM of 51% ± 6.6 (n = 760, n = 4 mice) of LPBN neurons positive for *Pdyn* were also positive for *Penk* labeling (*Figure 2C,K*). Reciprocally, a mean ± SEM of 58% ± 2.3 (n = 650 cells, n = 4 mice) of cells labeled by *Penk* probes were also labeled by *Pdyn* probes (*Figure 2D,K*). These FISH based

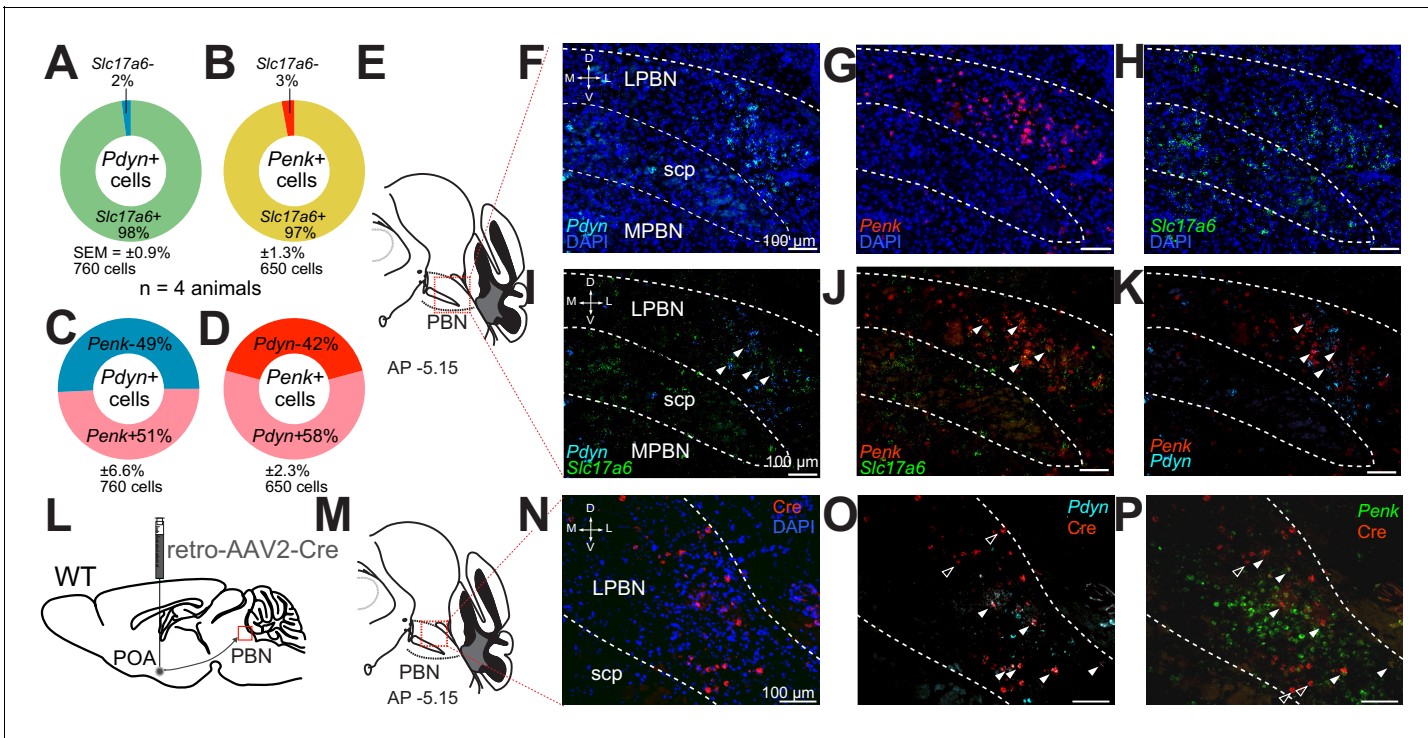

**Figure 2.** *Pdyn*+ and *Penk*+ LPBN neuron populations overlap, express *Slc17a6*, and project to the POA. (A–D) Quantification of cells labeled with (A) *Pdyn* probe (*Pdyn*+) and *Slc17a6* (VGLUT2+) probes, or (B) *Penk* (*Penk*+) and *Slc17a6* (VGLUT2+) probe, or (C and D) *Pdyn* and *Penk* probes. (E) Illustration of area of parabrachial nucleus (PBN) depicted in F–K. (F–H) Representative FISH images of LPBN neurons expressing (F) *Pdyn*, (G) *Penk*, and (H) *Slc17a6*. (I–K) (similar results were obtained in n = 3 mice) Representative images of overlays of (I) *Pdyn* with *Slc17a6* and *Penk* with *Slc17a6* (J), and (K) *Pdyn* with *Penk*. Arrowheads mark examples of cells positive for co-labeling of two transcripts. 98% of neurons expressing *Pdyn* and 97% of neurons labeled for *Penk* were also labeled with probes for *Slc17a6*. Data are presented as mean ± SEM; n = 4 animals, 760 cells for *Pdyn* and n = 4 animals, 760 cells for *Penk*. Diagram of viral injections into wild-type mice. (M) Anatomical location of representative FISH images shown in (N and O) that show overlap of (N) Cre expression, mediated by retrovirus transduction, with (O) *Pdyn* and (P) *Penk*. Arrowheads mark cells expressing Cre, with filled arrowheads co-expressing (O) *Pdyn* or (P) *Penk* and open arrowheads only expressing Cre.

The online version of this article includes the following figure supplement(s) for figure 2:

**Figure supplement 1.** POA-projecting parabrachial nucleus (PBN) neurons are VGLUT2+ and a subpopulation is Pdyn+.

**Figure supplement 2.** Expression of *Pdyn* and/or *Penk* in parabrachial nucleus (PBN)→POA projecting neurons and partially overlapping expression of *Cck* and *Pdyn* in LPBN.

experiments indicate that *Pdyn+* and *Penk+* cells in the LPBN express *Slc17a6* and are partially over-lapping subpopulations of glutamatergic LPBN cells. A recent report on PBN→POA neurons implicated cholecystokinin (Cck) expressing LPBN neurons in heat defense (*Yang et al., 2020*). We examined if *Pdyn* labeled neurons in LPBN were co-labeled by probes for *Cck* and found that 70% ± 0.7 (mean ± SEM, n = 150 cells, n = 3 mice) of LBP *Pdyn* labeled neurons were co-labeled by *Cck* probes (Supplemental *Figure 2—figure supplement 2D–F*) suggesting that mRNA for *Cck* and *pDyn* is expressed in overlapping neuronal populations.

Next, we examined the connection of *Pdyn* and *Penk* expressing neurons in the LPBN to the POA using retrograde AAVs and FISH. We injected AAV2-retro-Cre into POA of wild-type mice (*Figure 2L*) and collected brain sections containing LPBN. We probed these sections for viral induced Cre expression (*Figure 2M*). Using FISH, we observed retrograde viral induced expression of Cre in LPBN (*Figure 2N*) in cells also labeled with *Pdyn* (*Figure 2O*) and *Penk* (*Figure 2P*) indicating that neurons expressing these two opioid peptides project to the POA. To probe whether the PBN→POA neuronal population co-expresses *Pdyn* and/or *Penk*, we injected retro-AAV-Cre-GFP into the POA and probed LPBN containing brain sections with FISH probes for *GFP*, *Pdyn*, and *Penk*. We found that of *GFP* labeled cells in the LPBN, 49 ± 4% (mean ± SEM) were labeled by both *Penk* and *Pdyn* probes (*Figure 2—figure supplement 2A–C*). Of the remaining *GFP* labeled LPBN neurons, 26 ± 2% were labeled by *Pdyn* and 12 ± 1% by *Penk* (mean ± SEM). 13 ± 3% (mean ± SEM) of the quantified GFP labeled LPBN neurons were not labeled by either *Pdyn* or *Penk* probes (n = 3 mice, 169 cells).

To further examine the projections of Pdyn+ and Penk+ LPBN neurons to the POA, we employed both retrograde AAV's and anterograde tracing in Pdyn-Cre and Penk-Cre mice. To identify anterograde projections of LPBN neurons, we injected the Pdyn-Cre or Penk-Cre mice with AAV5-Ef1a-DIO-eYFP or AAV5-Ef1a-DIO-ChR2-eYFP into the LPBN. To retrogradely label POA projecting neurons we injected AAV2-retro-CAG-FLEX-tdTomato-WPRE into the POA of the same Pdyn-Cre or Penk-Cre animals (*Figure 3A,F*). In this experiment we observed anterograde labeling of processes with eYFP in the POA, from viral injections in the PBN, with dense projections in the ventral medial preoptic hypothalamus (VMPO) from both Pdyn-Cre (*Figure 3C*) and Penk-Cre (*Figure 3H*) mice. Retrograde labeling of LPBN neurons by Cre-dependent expression of tdTomato from retroAVV injected into the POA was evident in sections from both Pdyn-Cre (*Figure 3E*) and Penk-Cre (*Figure 3J*) brains. Double-labeled cells expressing both tdTomato (retrograde) and eYFP were present in the LPBN of both Pdyn-Cre and Penk-Cre mice (arrowheads in *Figure 3E and J*). In sagittal sections of brains taken from Pdyn-Cre mice injected with AAV-DIO-ChR2e-YFP in the PBN we also observed labeled projections to the POA among other brain areas (*Figure 2—figure supplement 1J,K*).

To examine which neurons comprise the PBN to POA projecting population, we injected mice expressing Cre under control of the VGLUT2 (*Slc17a6*) promoter (VGLUT2-Cre) (*Vong et al., 2011*) with AAV5-DIO-ChR2e-YFP bilaterally in the PBN, labeling VLGUT2 expressing PBN neurons (*Figure 2—figure supplement 1E*). We observed VGLUT2-Cre positive cells labeled by eYFP in the MPBN and LPBN after viral injection (*Figure 2—figure supplement 1F,G*). VGLUT2+ projections from the PBN to the POA including the VMPO and the median preoptic nucleus (MNPO) were labeled by AAV5-DIO-ChR2-eYFP injected in the PBN (*Figure 2—figure supplement 1H,I*). To determine whether Pdyn+ or VGLUT2+ cells represented the whole of the population of PBN to POA neurons, a retrograde recombinase dependent red-to-green (tdTomato to EGFP) Cre-switch virus (AAV-retro-DO_DIO-tdTomato_EGFP) was injected into the POA of Pdyn-Cre or VGLUT2-Cre mice (*Figure 2—figure supplement 1A*). In Pdyn=Cre mice, we observed cells in LPBN expressing tdTomato (Cre negative cells) and neurons expressing eGFP (Cre positive cells) (*Figure 2—figure supplement 1C*). In VGLUT2-Cre mice, we only observed eGFP expressing (Cre positive cells) neurons in LPBN (*Figure 2—figure supplement 1D*) indicating that the PBN to POA projection is composed entirely of VGLUT2+ cells. Taken together, results from FISH experiments and viral tracing studies indicate that Pdyn+ and Penk+ neurons in LPB project to the POA, particularly the VMPO, and that both Pdyn+ and Penk+ POA projecting neurons are subsets of the VGLUT2+ population of LPBN neurons that project to POA.

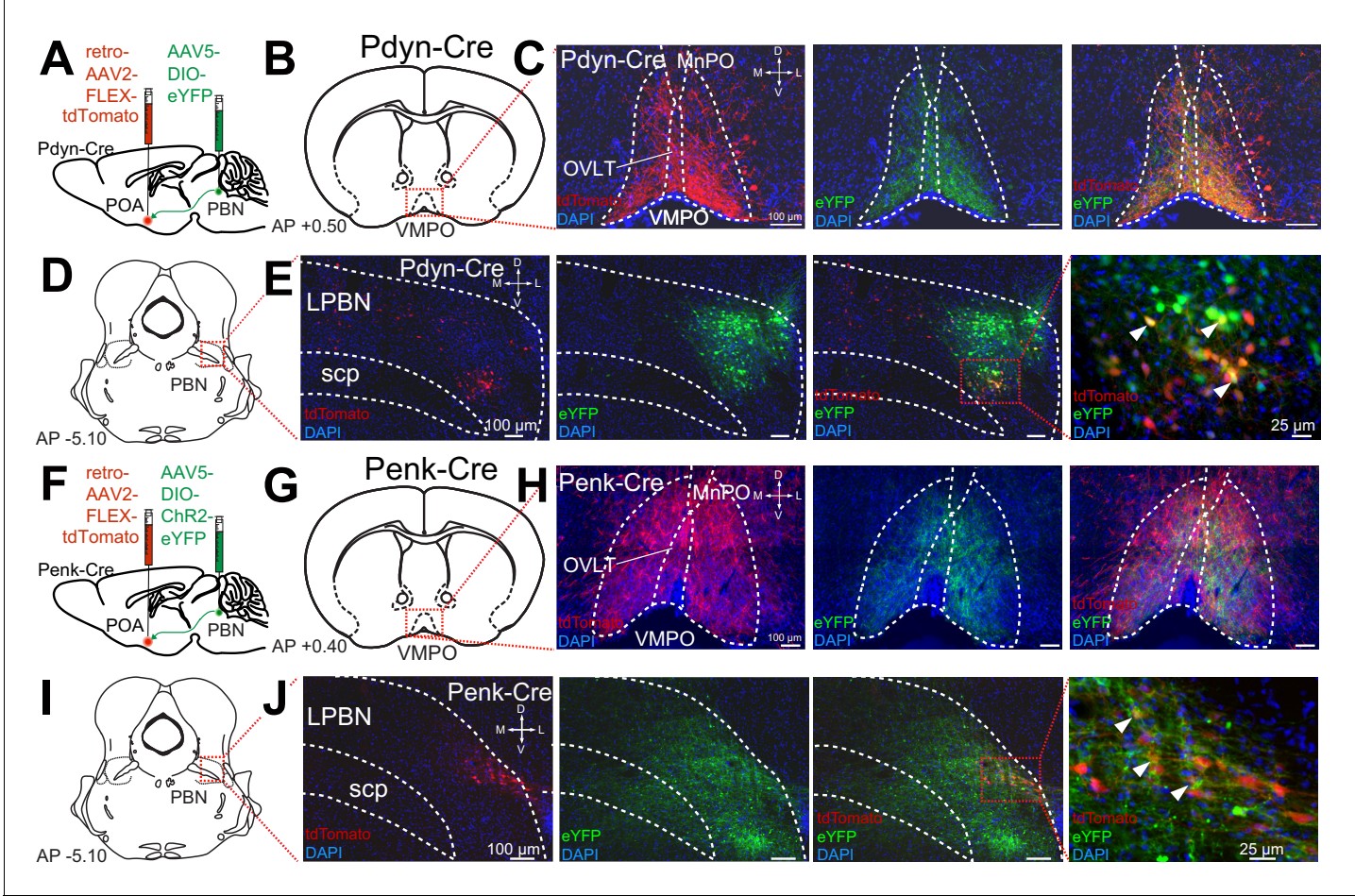

**Figure 3.** Pdyn+ and Penk+ LPBN neurons project to VMPO. (**A**) Illustration of injection of retroAAV-DIO-tdTomato in POA and AAV5-DIO-eYFP in a Pdyn-Cre mouse. (**B**) Diagram of POA region depicted in (**C**) showing antero- (green) and retrograde (red) labeling of Pdyn+ neurons in POA. (**D**) Diagram of parabrachial nucleus (PBN) region depicted in (**E**) showing retrograde labeling from POA (red) and eYFP expression (green). Yellow cells in overlay image, marked with arrow heads, illustrate dual labeling by locally injected and retrograde viruses. (**F**) Illustration of injection of retroAAV-DIO-tdTomato in POA and AAV5-DIO-eYFP in an Penk-Cre mouse. (**G**) Diagram of POA region depicted in (**H**) show antero- (green) and retrograde (red) labeling of Penk+ neurons in POA. (**I**) Diagram of PBN region depicted in (**J**) showing retrograde labeling from POA (red) and eYFP expression (green). Yellow cells in overlay image, marked with arrow heads, illustrate dual labeling by locally injected and retrograde viruses.

## Photostimulation of Pdyn$^{PBN \rightarrow POA}$, Penk$^{PBN \rightarrow POA}$, and VGLUT2$^{PBN \rightarrow POA}$ generates rapid onset of hypothermia

Using the respective Cre driver lines, we next examined the roles of POA-projecting Pdyn+, Penk+, and VGLUT2+ PBN neurons (circuits are denoted as Pdyn$^{PBN \rightarrow POA}$, Penk$^{PBN \rightarrow POA}$, and VGLUT2$^{PBN \rightarrow POA}$, respectively) in regulating body temperature. We injected AAV5-DIO-ChR2-eYFP bilaterally into the LPBN of Pdyn-Cre, Penk-Cre, and VGLUT2-Cre mice, and after 6 weeks, we implanted a single midline optic fiber above VMPO, where projections from PBN were observed (*Figure 4A,B*). We implanted mice with a subdermal wireless temperature transponder to enable touch free recording of body temperature. For each trial, we connected mice to an optic patch cable, and following a 1-hr period of habituation to the behavioral arena, we photostimulated PBN→POA terminals for 15 min with 10 ms light pluses at pulse frequencies of 2, 5, 10, and 15 Hz (*Figure 4C*). We recorded body temperature every 5 min for 65 min, beginning 5 min prior to photostimulation (*Figure 4C*).

Photostimulation of Pdyn$^{PBN \rightarrow POA}$ neuron terminals caused rapid and significant reduction in body temperature in Pdyn-Cre mice (n = 6), with increasing magnitude of drop in body temperature corresponding to increasing photostimulation frequency up to 10 Hz (*Figure 4D*). 15 min of

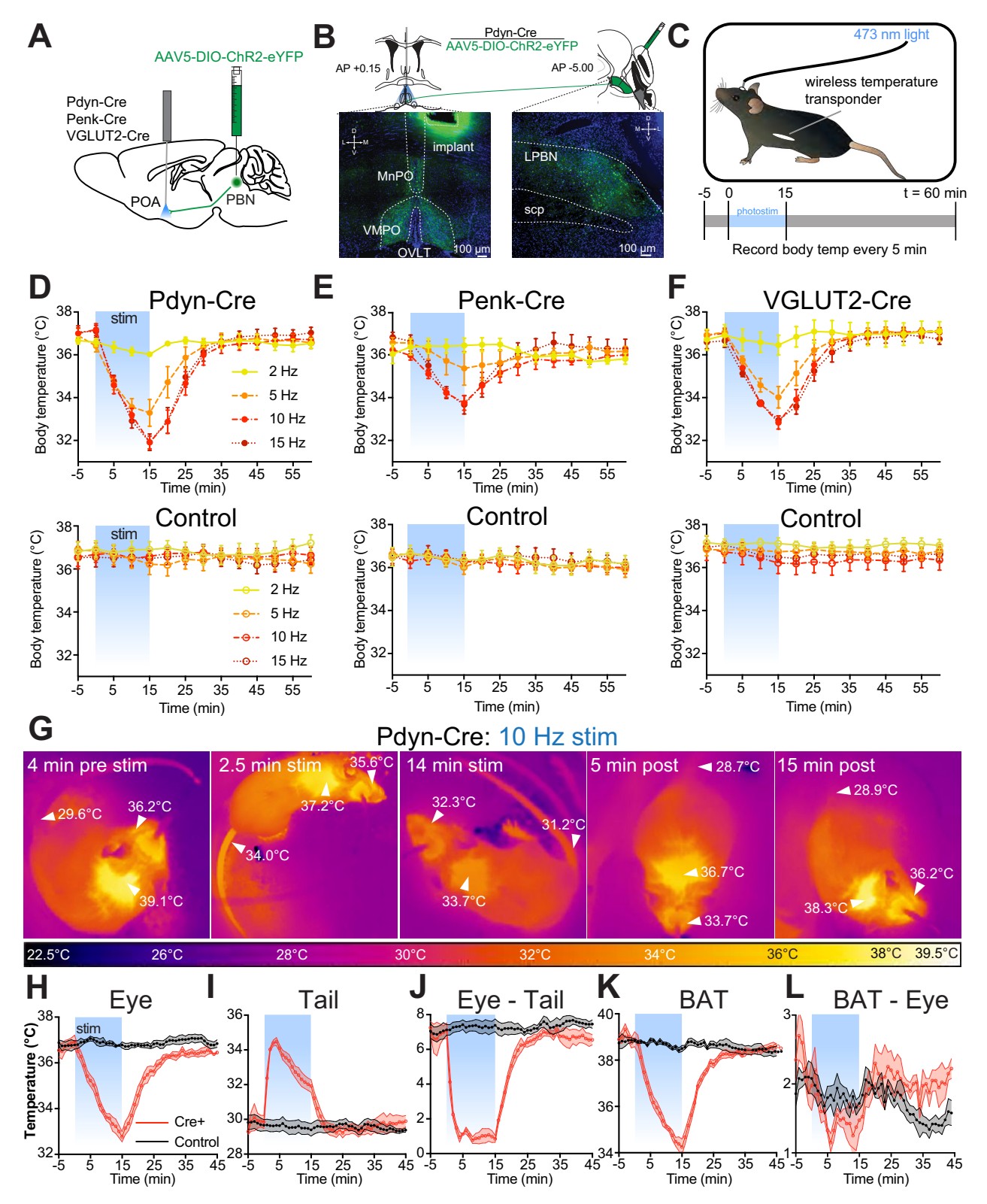

**Figure 4.** Photostimulation of Pdyn$^{PBN \rightarrow POA}$, Penk$^{PBN \rightarrow POA}$, and VGLUT2$^{PBN \rightarrow POA}$ causes acute hypothermia by evoking thermal heat defenses. (**A**) Illustration of viral injections in parabrachial nucleus (PBN) and fiber optic implantation over POA in Pdyn-Cre, Penk-Cre, or VGLUT2-Cre mice. (**B**) Illustration shows viral and fiber optic delivery in a Pdyn-Cre mouse along with representative expression of ChR2-eYFP (green) in PBN injection site and POA implantation site. (**C**) Diagram shows core body temperature measurement method and paradigm for photostimulation for 15 min and

*Figure 4 continued on next page*

*Figure 4 continued*

temperature recording for 65 min trials. (D–F) Body temperature vs. time graphs for 2 (yellow), 5 (orange), 10 (red), and 15 (dark red) Hz photostimulation of (D) Pdyn$^{PBN \to POA}$, (E) Penk$^{PBN \to POA}$, (F) VGLUT2$^{PBN \to POA}$, and controls for each. Photostimulation was delivered from t = 0 to t = 15 min and led to a frequency dependent reduction in body temperature in Pdyn-Cre, Penk-Cre, and VGLUT2-Cre mice. Body temperature of control animals was stable throughout the trials. Data are presented as mean ± SEM. For experimental animals, n = 6 (D and E) and n = 8 (F). For control animals, n = 8 (D) and n = 7 (E and F). (G) Representative quantitative thermal imaging from a representative trial showing a mouse before, during, and after 10 Hz photostimulation of Pdyn$^{PBN \to POA}$. Arrows show temperatures of eye, BAT, or tail. Eye and BAT temperature decreased as a result of stimulation; tail temperature increased as a result of stimulation. (H–L) Quantitative thermal imaging measurements of (H) eye, (I) tail, (J) eye minus tail, (K) BAT, and (L) BAT minus eye temperature vs. time graphs for 10 Hz photostimulation of Pdyn$^{PBN \to POA}$. Photostimulation was delivered from t = 0 to t = 15 min and led to decreases in eye and BAT temperatures, an increase in tail temperature. Tail and eye temperatures equilibrated in Cre+ animals. BAT thermogenesis was suppressed with a decline in the difference between eye and BAT temperatures during stimulation. Data are presented as mean ± SEM. See *Figure 4—figure supplement 1* for data from Penk-Cre animals.

The online version of this article includes the following figure supplement(s) for figure 4:

**Figure supplement 1.** Pdyn$^{PBN \to POA}$ or Penk$^{PBN \to POA}$ photostimulation-induced hypothermia is independent of opioid system.

stimulation of Pdyn$^{PBN \to POA}$ projections reduced the body temperature to 36.0 ± 0.1°C at 2 Hz (p=0.571 vs. control), 33.3 ± 0.6°C (p=0.0032) at 5 Hz, 31.9 ± 0.3°C, (p<0.0001) at 10 Hz, and 31.9 ± 0.4°C (p<0.0001) at 15 Hz compared to control. In control mice (n = 7), photostimulation did not cause significant changes in body temperature at any of the tested frequencies (*Figure 4D*).

Photostimulation of Penk$^{PBN \to POA}$ neuron terminals also caused a rapid reduction in body temperature (*Figure 4E*) in a stimulation frequency dependent manner. 15 min of stimulation of Penk$^{PBN \to POA}$ projections in Penk-Cre mice reduced body temperature to 36.4 ± 0.3°C at 2 Hz (p=0.999 vs. control), 34.9 ± 0.7°C (p=0.495) at 5 Hz, 33.8 ± 0.3°C (p=0.0001) at 10 Hz, and 33.7 ± 0.4°C (p=0.0002) at 15 Hz, compared to a separate cohort of control mice (n = 7) which did not display altered body temperatures in response to photostimulation.

In VGLUT2-Cre mice with AAV-DIO-ChR2-eYFP injected into PBN, stimulation of VGLUT2$^{PBN \to POA}$ terminals in POA also caused a rapid and significant decrease in body temperature (*Figure 4F*). 15 min of photostimulation in VGLUT2-Cre mice (n = 8) significantly reduced the mean ± SEM body temperature to 36.5 ± 0.5°C at 2 Hz (p=0.257 vs. control) 34.0 ± 0.5°C at 5 Hz (p=0.0005), 32.8 ± 0.3°C (p<0.0001) at 10 Hz, and 32.9 ± 0.2°C (p<0.0001) at 15 Hz compared to control mice (n = 7). The average changes in body temperature that we measured in Pdyn-Cre and VGLUT2-Cre mice were not significantly different at any of the tested stimulation frequencies. The body temperature reduction evoked by photostimulation in Penk-Cre mice was smaller in magnitude than that in either Pdyn-Cre or VGLUT-Cre mice. The mean body temperature we measured in Penk-Cre mice after 15 min of simulation was significantly different than Pdyn-Cre at 10 Hz (p=0.02), with activation of the Penk+ terminals having less of an effect. These data demonstrate that activation of PBN→POA terminals causes rapid decreases in body temperature.

## Photostimulation of Pdyn$^{PBN \to POA}$ and Penk$^{PBN \to POA}$ terminals causes vasodilation and suppresses brown fat thermogenesis

We sought to examine mechanisms causing core body temperature reduction in response to photostimulation of PBN→POA projections. We used thermal imaging to measure temperatures of eye, tail, and interscapular region, which covers BAT, in Pdyn-Cre mice (representative imaging in *Figure 4G*). Thermal imaging of the eye has previously been demonstrated as an accurate proxy for core body temperature (*Vogel et al., 2016*). We recorded eye temperatures every minute during a 10 Hz photostimulation paradigm, as described above. Recorded eye temperatures demonstrated a rapid reversible decrease after photostimulation (*Figure 4H*) and closely tracked values obtained using implanted wireless transponders. In Pdyn-Cre mice, mean ± SEM eye temperature dropped from 36.9°C ± 0.3 to 32.8°C ± 0.2 with 15 min of stimulation (*Figure 4H*). Thermal imaging to quantify tail temperature can be used to observe heat loss from vasodilation in response to warmth (*Meyer et al., 2017*). We obtained thermal imaging measurements of the tail temperatures approximately 1 cm from the base of the tail each minute. In Pdyn-Cre mice, tail temperature measurements demonstrated a very rapid increase following the onset of photostimulation, increasing a mean ± SEM of 4.2°C ± 0.5 after 2 min of photostimulation (*Figure 4I*). Increase in tail temperature preceded the decline in core body temperature. As core body temperature began to decrease, the tail

temperature also began to decrease (*Figure 4H and I*). We examined the difference between the tail and eye temperatures (*Figure 4H–J*) to determine whether the gradient between core and peripheral temperature was maintained as body temperatures declined during stimulation. At baseline we observed a mean ± SEM difference 6.9 ± 0.29°C between the measured eye and tail temperatures. Eye–tail temperature difference significantly (p<0.0001) decreased compared to control to a mean ± SEM of 1.3 ± 0.3°C and remained stable during photostimulation even as body temperature declined. The difference in eye–tail temperature returned to baseline shortly after photostimulation was stopped (*Figure 4J*).

Previous studies have implicated the POA in regulating BAT activation in response to cooling (*Nakamura and Morrison, 2007*; *Tan et al., 2016*). To simultaneously examine changes in BAT activity in response to the PBN→POA photostimulation-induced hypothermia, temperature measurements were also made of the interscapular BAT region temperature in mice with the fur removed from over the intrascapular region. In Pdyn-Cre mice, the temperature of the BAT region decreased rapidly following the onset of stimulation and returned to baseline post-stimulation in a pattern similar to body temperature (*Figure 4K*). If BAT activity responded to the decrease in body temperature by increasing metabolism, then the BAT–eye temperature difference would be expected to increase, reflecting the warming activity of BAT and the falling body temperature. The temperature difference between BAT and eye (BAT–eye) decreased during the period of stimulation but returned to baseline when stimulation was stopped (*Figure 4L*). We conducted similar experiments using thermal imaging in Penk-Cre and additional control animals (*Figure 4—figure supplement 1*). In the Penk-Cre mice, we found similar effects, but of smaller overall amplitude. Photostimulation of Penk$^{PBN→POA}$ terminals led to a decrease in eye temperature, a rapid increase in tail temperature, decrease in BAT temperature, and collapse of the eye–tail temperature gradient (*Figure 4—figure supplement 1G–J*). Together these results indicate that PBN to POA neurons can drive physiologic adaptation to lower body temperature by increasing heat dissipation and suppressing thermogenesis.

## Changes in body temperature evoked by photostimulation of Pdyn$^{PBN→POA}$ and Penk$^{PBN→POA}$ terminals are opioid peptide and receptor independent

To test the potential role of endogenous opioids and their receptors in mediating the alterations in body temperature evoked by activation of Pdyn$^{PBN→POA}$ and Penk$^{PBN→POA}$ terminals, mice were treated with opioid receptor antagonists prior to photostimulation (*Figure 4—figure supplement 1A*). Pdyn-Cre (n = 7) and control mice (n = 7) were treated with the opioid receptor antagonist naltrexone (3 mg/kg) via intraperitoneal (IP) injection and then given a 10 Hz photostimulation paradigm as above (*Figure 4—figure supplement 1B*). The order of naltrexone and saline was varied between animals, and trials were run on separate days. 30 min after treatment with naltrexone, we did not observe a significant impact on photostimulation induced change in body temperature compared to saline treated animals. Naltrexone was paired with the Pdyn-Cre line because of the relatively higher affinity of naltrexone for kappa opioid receptors compared to naloxone (*Meng et al., 1993*). A distinct cohort of Pdyn-Cre mice (n = 4) was treated with saline and for subsequent trials with Norbinaltorphimine (norBNI) 10 mg/kg via IP injection 1 day prior and again 30 min prior to photostimulation. Pretreatment with norBNI did not significantly alter the decrease in body temperature induced by 10 Hz photostimulation of Pdyn$^{PBN→POA}$ terminals (*Figure 4—figure supplement 1B*). We confirmed that the doses and time courses of our naltrexone and norBNI administration were effective by examining the block of suppression of locomotion by the kappa receptor agonist U50 by the antagonists naltrexone or norBNI (*Paris et al., 2011*). Pretreatment with naltrexone or norBNI mitigated U50 mediated suppression of locomotor activity (*Figure 4—figure supplement 1D–F*). Penk-Cre mice (n = 5) injected with AAV5-DIO-ChR2eYFP in the PBN and control mice (n = 4) were treated with naloxone 5 mg/kg and saline, with the order of treatments varied between animals and trials conducted on separate days. 30 min after treatment with naloxone, Penk$^{PBN→POA}$ terminals were photostimulated at 10 Hz. No significant effect of naloxone on photostimulation-induced changes in body temperature was observed (*Figure 4—figure supplement 1*). These data suggest that the acute alterations in body temperature due to stimulation of Pdyn$^{PBN→POA}$ and Penk$^{PBN→POA}$ terminals are not driven by endogenous opioid release and subsequent opioid receptor signaling.

## Glutamatergic PBN neuronal activity is necessary for heat-induced vasodilation

Glutamatergic signaling in the POA and in PBN has previously been implicated in heat defensive behaviors (*Nakamura and Morrison, 2010*), and our results, presented here, demonstrate sufficiency of PBN VGLUT2+ neurons in driving hypothermia (*Figure 4F*). To examine the necessity of VGLUT2+ PBN neurons in mediating heat defensive behaviors in awake behaving animals, AAVs encoding Gi coupled DREADDs in a Cre-dependent manner (AAV-DIO-hM4DGi) were injected into PBN bilaterally in VGLUT2-Cre mice (n = 5). Mice were treated with saline or clozapine-N-oxide (CNO) (2.5 mg/kg IP) 30 min prior to a heat challenge of 34°C for 15 min and were tested with the reciprocal during a subsequent trial more than 24 hr later. We used a custom small arena with floor and walls lined with a water jacket connected to circulating water baths at 20°C or 34°C to create a rapid change in temperature between two stable set points while allowing for continuous thermal imaging (*Figure 5A,B*). Using quantitative thermal imaging, we measured tail temperatures and arena floor temperatures (depicted by the yellow-orange shaded areas) every minute during chemogenetic inhibition of VGLUT2 activity (*Figure 5B–D*). In mice treated with saline, measurements of tail temperatures showed a rapid rise following the shift of arena temperature to 34°C and measured tail temperatures were higher than the arena floor temperature (*Figure 5B,D, and F*). In mice treated with CNO, which activated the inhibitory DREADD in VGLUT2+ PBN neurons, the mean ± SEM tail temperature after 15 min of exposure to 34°C was 34.8°C ± 0.5, significantly (p=0.01) lower than the corresponding average tail temperature after saline treatment, 37.1°C ± 0.5 (*Figure 5D*). The tail temperature in saline treated mice exceeded the temperature of the arena floor (*Figure 5B and F*), but in CNO treated mice, tail temperature rose only to the temperature of the floor (*Figure 5B and E*). Consistent with an effect of passive heating of the tail, as opposed to the active vasodilation evoked by the thermal challenge, the rate of increase in the tail temperature was also slower following CNO treatment compared to saline (*Figure 5C*). After return of the arena floor temperature to 20°C, tail temperatures returned to a baseline of approximately 22°C following both saline and CNO treatments. Similar experiments carried out in Pdyn-Cre mice demonstrated that Gi DREADD mediated inhibition of Pdyn+ PBN neurons is not sufficient to prevent vasodilation in response to thermal heat challenge (*Figure 5—figure supplement 1*). CNO treatment in WT mice had no significant effects on tail temperature changes compared to saline treatment (*Figure 5—figure supplement 1*). These results indicate that VGLUT2+ PBN neurons are required for heat defensive responses including physiological vasodilation.

## Photostimulation of PBN→POA drives thermal defensive behaviors

The PBN has been found to play essential roles in driving escape and aversive learning to nociceptive stimuli. Previous studies have shown that the spinothalamic pathway is not required for behavioral thermoregulation and that muscimol mediated inhibition of PBN blocked thermal preference seeking (*Yahiro et al., 2017*). To test the sufficiency of Pdyn[PBN→POA], Penk[PBN→POA], and VGLUT2[PBN→POA] to drive avoidance behavior, we conducted real-time place aversion (RTPA) experiments using the respective Cre driver lines and photostimulation of terminals in the POA. Photostimulation of terminals was paired to entry into one compartment of a balanced two-compartment conditioning apparatus void of salient stimuli. Neurons that encode a negative valence will cause an aversion from the chamber paired with photostimulation, and those with a positive valence will drive a preference for it (*Kim et al., 2013*; *Namburi et al., 2016*; *Siuda et al., 2015*; *Stamatakis and Stuber, 2012*; *Tan et al., 2012*). As in experiments above, we injected AAV-DIO-ChR2-eYFP into the PBN bilaterally of Cre driver line mice and implanted optical fibers over the POA (*Figure 6A,D, G*). Photostimulation of Pdyn[PBN→POA] terminals drove aversion in a frequency dependent manner, with time spent in the stimulation side being significantly lower (p<0.0001) at 5, 10, and 20 Hz stimulation frequencies compared to control mice (*Figure 6A–C*). Results from parallel RTPA experiments using Penk-Cre mice demonstrated a similar effect of aversion seen at 5 (p=0.0002), 10 (p<0.0001), and 20 Hz (p<0.0001) stimulation frequencies compared to control mice (*Figure 6D–F*). Results we obtained using VGLUT2-Cre mice in RTPA experiments showed significant (p<0.0001) decreases in time spent on the stimulation side at 2, 5, 10, and 20 Hz compared to control animals (*Figure 6G–I*). For each genetic line, we examined the locomotor activity or distance traveled during the RTPA. In Pdyn-Cre mice, we observed a small but significant (p=0.009) difference in mean ± SEM total

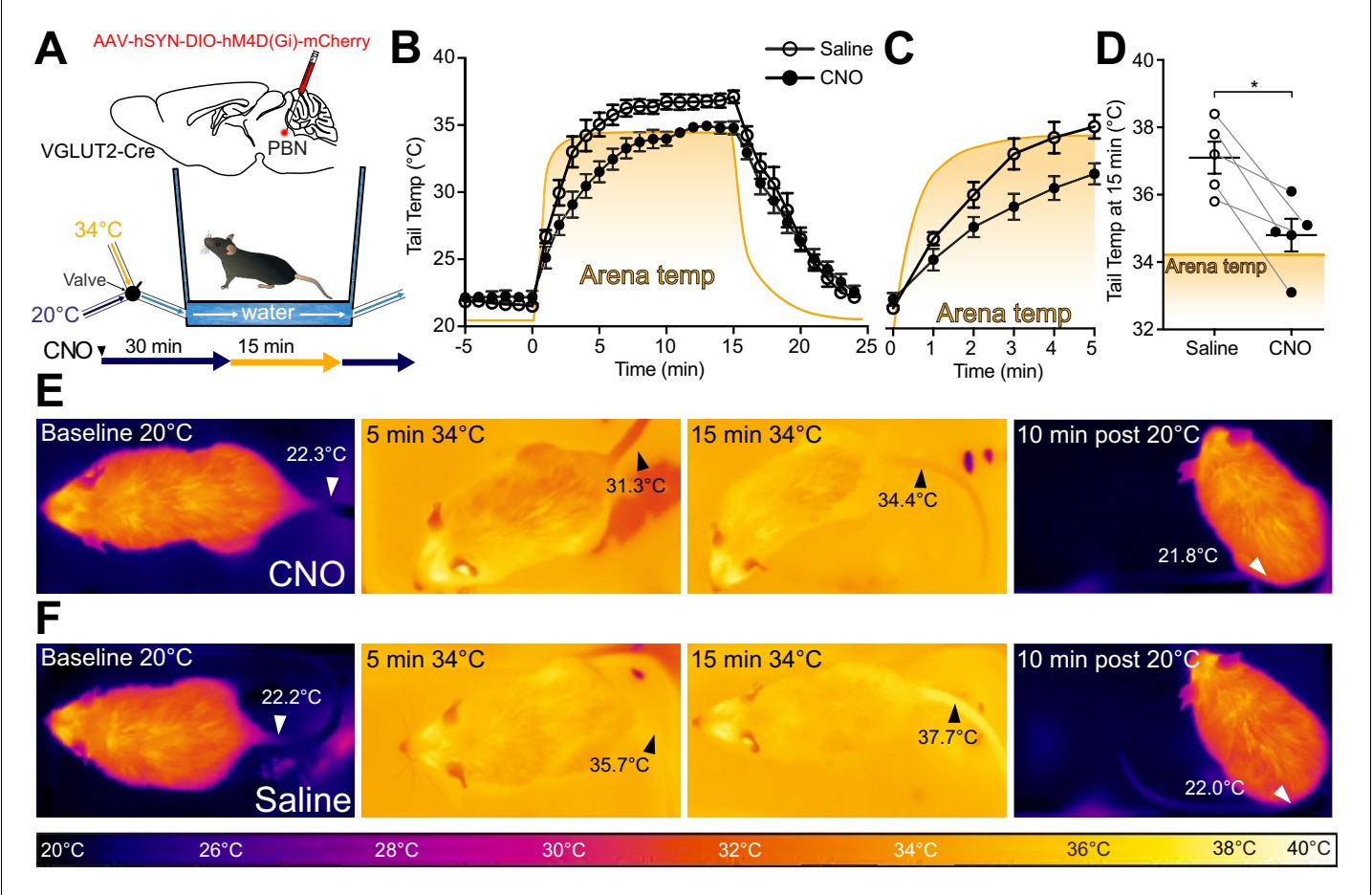

**Figure 5.** VGLUT2+ parabrachial nucleus (PBN) neurons are necessary for heat-defensive tail vasodilation. (**A**) Illustrations depict viral injections in VGLUT2-Cre mice and purpose-built heat challenge arena that allowed for rapid changing of environmental temperature between two stable set points. (**B**) Tail temperature as determined using quantitative thermal imaging vs. time graph for 34°C thermal heat challenge for mice expressing hM4D (Gi) DREADDs in VGLUT2+ PBN neurons treated either with CNO or saline. Heat challenge was delivered from t = 0 to t = 15 min, and arena temperature measured using thermal imaging during the trial is represented by the orange line. In mice injected with CNO 2.5 mg/kg, tail temperature passively equilibrated with arena temperature (34°C) over the 15 min heat challenge. In mice injected with saline, tail temperature rose above arena temperature after 5 min of heat challenge representing heat release through vasodilation. Data are presented as mean ± SEM. n = 5 animals, paired between CNO and saline conditions. (**C**) Tail temperature vs. time graph for 34°C heat challenge between t = 0 and t = 5 min. Note the separation between average tail temperatures of the saline condition vs. the CNO condition. Data are presented as mean ± SEM. n = 5 animals, paired between CNO and saline conditions. (**D**) Tail temperature at t = 15 min of 34°C heat challenge. Tail temperatures in the saline condition were an average of 2.3 ± 0.68°C higher than those in the CNO condition. (**E**) Representative thermal images of trials for mice treated with CNO and measurement of tail temperature showing tail temperatures remain close to the temperature of the area floor. (**F**) Representative thermal images of trials for mice treated with saline and tail temperature exceed floor temperature. Data are presented as mean ± SEM. n = 5 animals, paired between CNO and saline conditions. Student's t-test, *p<0.05. See *Figure 5—figure supplement 1* for data from the same assay in Pdyn-Cre mice.

The online version of this article includes the following figure supplement(s) for figure 5:

**Figure supplement 1.** Gi DREADD mediated inhibition of Pdyn+ parabrachial nucleus (PBN) neurons does not block thermal challenge evoked tail vasodilation and CNO in WT mice does not alter responses to warmth challenge.

distance traveled only during trials using 20 Hz stimulation – 29 m ± 2 (n = 8) compared to control 48 m ± 5 (n = 7) – but not during trials using lower stimulation frequencies (*Figure 6—figure supplement 1A*). In Penk-Cre and VGLUT2-Cre mice, we observed no significant differences between Cre+ and control animals at any of the photostimulation frequencies (*Figure 6—figure supplement 1B, C*). Comparisons of male vs female mice did not reveal sex dependent effects in the acute hypothermic changes in body temperature evoked by photostimulation of PBN→POA terminals (*Figure 6— figure supplement 1D,E*).

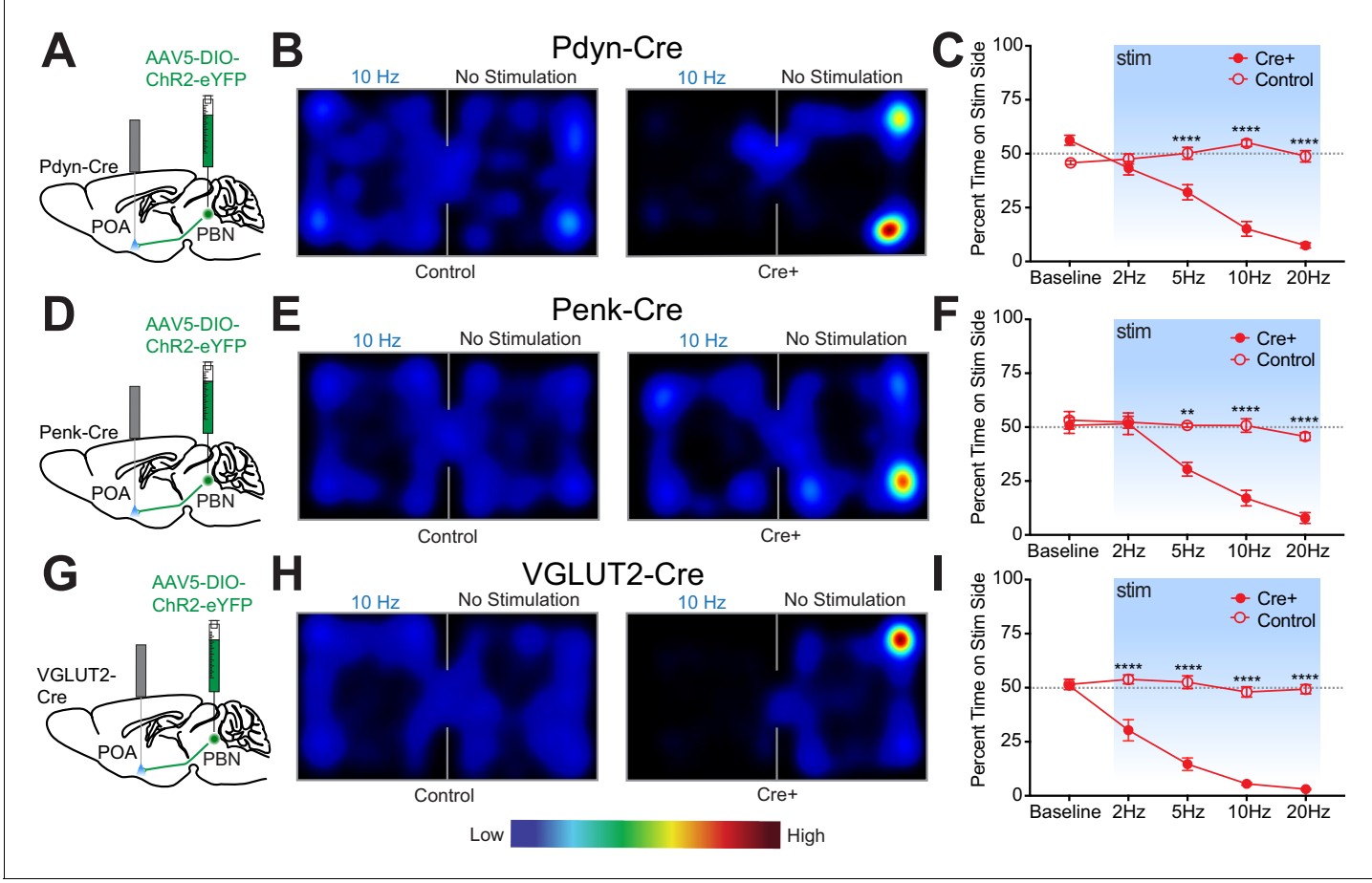

**Figure 6.** Photostimulation of Pdyn[PBN→POA], Penk[PBN→POA], and VGLUT2[PBN→POA] terminals induces real time place aversion. (**A, D, and G**) Illustrations of viral injections in parabrachial nucleus (PBN) and fiber optic implantations over POA in Pdyn-Cre mice, Penk-Cre, and VGLUT2-Cre mice, respectively. (**B, E, and H**) Representative heat maps showing spatial distribution of time-spent behavior resulting from side-conditional 10 Hz photostimulation of control or Pdyn-Cre, Penk-Cre, and VGLUT2-Cre mice, respectively. (**C**) For Pdyn-Cre vs control mice, frequency response of RTPP at 0 (baseline), 2, 5, 10, and 20 Hz. Data are presented as mean ± SEM; n = 6 Cre+, eight control; two-Way ANOVA, Bonferroni post hoc. (**F**) Penk-Cre frequency response of RTPP at 0 (baseline), 2, 5, 10, and 20 Hz. Data are presented as mean ± SEM; n = 6 Cre+, seven control; two-Way ANOVA, Bonferroni post hoc (5 Hz ChR2 vs. 5 Hz control ***p<0.001, 10 Hz ChR2 vs. 10 Hz control ****p<0.0001, 20 Hz ChR2 vs. 20 Hz control ****p<0.0001). (**I**) VGLUT2-Cre frequency response of RTPP at 0 (baseline), 2, 5, 10, and 20 Hz. Data are presented as mean ± SEM; n = 8 Cre+, seven control; two-Way ANOVA, Bonferroni post hoc (2 Hz ChR2 vs. 20 Hz control ****p<0.0001, 5 Hz ChR2 vs. 5 Hz control ****p<0.0001, 10 Hz ChR2 vs. 10 Hz control ****p<0.0001, 20 Hz ChR2 vs. 20 Hz control ****p<0.0001). See also *Figure 6—figure supplement 1*.

The online version of this article includes the following figure supplement(s) for figure 6:

**Figure supplement 1.** Total distance traveled for Pdyn-Cre, Penk-Cre, and VGLUT2-Cre mice in real-time place aversion assay and male vs. female photostimulation-induced body temperature change in Pdyn-Cre and VGLUT2-Cre mice.

Other thermoregulatory behaviors including posture, stance, and locomotion are altered by exposure to warm temperatures (*Cabanac, 1975*). Therefore, we next examined alterations in locomotion using 20 min open field-testing trials in Pdyn-Cre and control mice (*Figure 7A–C*). Stimulation of Pdyn[PBN→POA] terminals at 10 Hz resulted in a large and significant (p=0.0008) decrease in mean ± SEM distance traveled: 26.1 m ± 6.2 (n = 5) in Pdyn-Cre compared to 65.9 m ± 5.6 in control mice (n = 7) (*Figure 7B,C*). Postural extension, depicted in the photograph (*Figure 7D*), a heat evoked behavior in rodents that reduces heat production by postural tone and increases exposed body surface to promote thermal transfer (*Roberts, 1988*), was evoked by photostimulation of Pdyn[PBN→POA] terminals. Scoring of video recordings of trials of Pdyn-Cre (n = 7) and control mice (n = 4) revealed that Pdyn-Cre mice quickly transition to a sprawled posture after the onset of 10 Hz photostimulation, which also induces hypothermia. Following the end of photostimulation, mice transition and spend more time in a posture with their tail curled under their bodies to minimize exposed surface

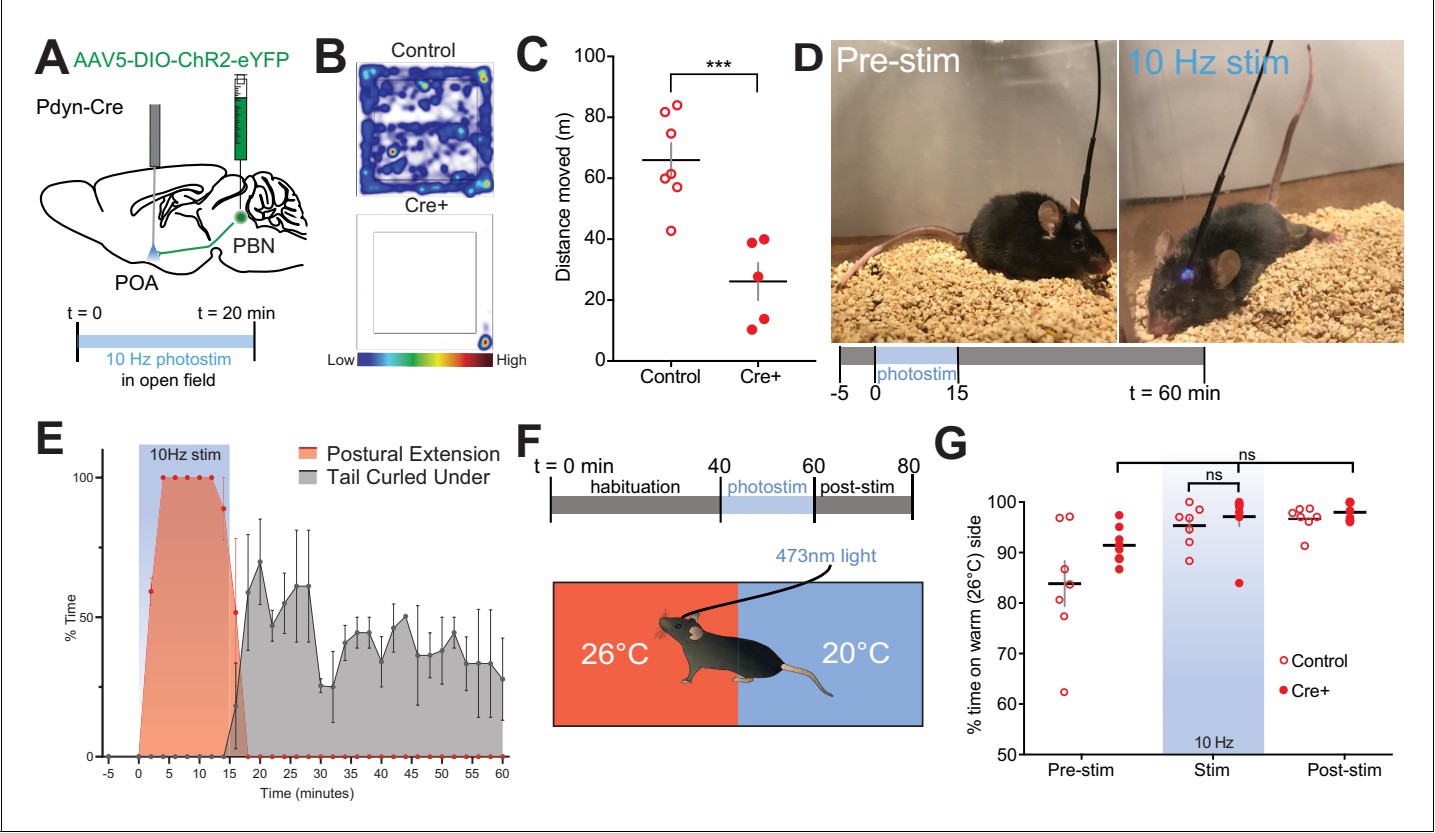

**Figure 7.** Photostimulation of Pdyn^PBN→POA suppresses locomotion, evokes postural extension but does not alter temperature preference. (A) Illustration of injection in parabrachial nucleus (PBN) and fiber implantation over POA in Pdyn-Cre mice. (B) Representative heat maps show spatial distribution of time-spent behavior resulting from constant 20 min 10 Hz photostimulation of control or Pdyn^PBN→POA. (C) Quantification of movement during open field testing. Control animals moved an average of 39.84 ± 8.33 meters more than Cre+ animals during open field trials. Data are presented as mean ± SEM; n = 5 Cre+, seven control; Student's t test, ***p<0.001 (D) 10 Hz photostimulation of Pdyn^PBN→POA leads to postural extension behavior as shown. Representative images of a mouse pre stimulation and during 10 Hz photostimulation of Pdyn^PBN→POA. (E) Quantification of percent time spent in time spent engaged in postural extension in Pdyn-Cre mice in two min time bins. Following onset of photostimulation mice engaged in postural extension (red). With termination of stimulation mice, we noted to switch to a posture with their tails curled under their bodies (grey). Postural extension was not observed in any control mice. (F) Overview of paradigm with three epochs: 40 min of pre-stim, 10 Hz photostimulation for 20 min, and post-stim for 20 min in an arena with aluminum floor held at 20°C and 26°C on opposing sides. (G) Quantification of time spent in each temperature area showed non-significant changes in percent time spent in each area during delivery of stimulation, with a strong preference for the 26°C side during all epochs. Data presented as mean ± SEM with individual values, n = 9 Pdyn-Cre (ANOVA ns = 0.7341 for Pdyn-Cre mice across epochs) and (t-test ns p>0.99 for Pdyn-Cre vs Control during stimulation epoch).

area (*Figure 7E*). We did not observe postural extension at any time in control mice during these trials.

Temperature selection is an important complex thermal defense behavior. Moving to an area with cooler environmental temperature, when possible, is a way to defend against excessive heat. Available studies indicate thermal selection requires the engagement of multiple poorly understood neural circuits. We next tested whether photoactivation of Pdyn^PBN→POA terminals is sufficient to induce a shift to slightly cooler temperature preference. To examine temperature preference, we placed mice in an arena with an aluminum floor in which each side is held at a set temperature of 20°C or 26°C (*Figure 7F*). Mice were habituated to the arena prior to the start of the trial to familiarize the animals to area. Trials consisted of a 40 min pre-stimulation period, a 20 min stimulation period, and a 20 min post-stimulation period. As expected, at baseline, mice spent a greater amount of time on the 26°C side (*Figure 7G*). In Pdyn-Cre mice (n = 6), photostimulation of the POA at 10 Hz did not alter animals' thermal preference to the cooler side of arena (*Figure 7G*), despite this photo-stimulation paradigm evoking hypothermia (*Figure 4D*) and driving other thermal defense behaviors.

This result indicates that the Pdyn$^{PBN\rightarrow POA}$ neurons are not sufficient to drive cool seeking behavior when activated, suggesting that other neural pathways are also likely required to drive this behavior.

## Discussion

In the present study we demonstrate that warm-activated neurons within the PBN overlap with neural populations (Pdyn+ and Penk+) marked by Cre reporters for expression of *Pdyn* and *Penk* (*Figure 1* and *Figure 2—figure supplement 2*). Employing FISH, we found that *Pdyn* and *Penk* expressing neuronal populations are glutamatergic (express *Slc17a6*) and partially overlap with each other (*Figure 2*). Using anterograde and retrograde viral tools, we demonstrate that Pdyn+, Penk+, and VGLUT2+ PBN neurons project to the POA (*Figure 3* and *Figure 2—figure supplement 1*). We found that photoactivation activation of terminals from Pdyn+ or Penk+ or VGLUT2+ PBN→POA drove physiological and behavioral heat defense behaviors (*Figure 8*).

### Overlapping populations of warm-activated PBN neurons express Penk and Pdyn, are glutamatergic, and project to the POA

We report that PBN neurons are activated following exposure to warmth. The majority of warm-activated PBN neurons are Pdyn+ and, surprisingly, a smaller population of these warm-activated PBN cells are Penk+ (*Figure 1*). 81% and 54% of cFos+ cells are Pdyn+ or Penk+, respectively, suggesting

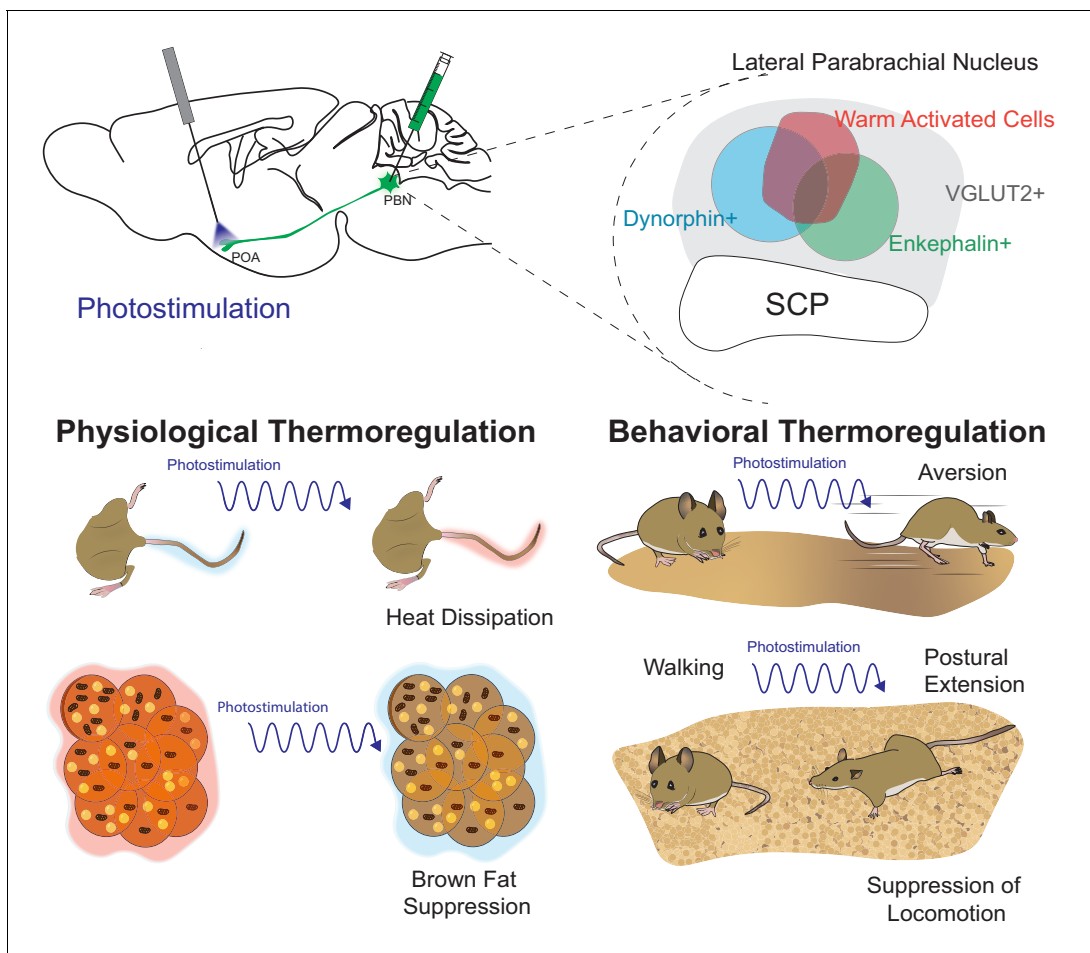

**Figure 8.** Graphical summary. The presented studies focused on parabrachial nucleus (PBN)→POA projecting cells by photostimulating terminals in the POA. We identified warm-activated neurons (red circle) in the lateral PBN that incompletely overlap with Penk+ and Pdyn+ PBN neuronal populations. Further, we found that these Penk (green circle) and Pdyn (blue circle) neurons express VGLUT2 (gray) and partially overlap with each other. Photostimulation of PBN→POA projections revealed that PBN VGLUT2+, Pdyn+, or Penk+ projections drive physiological and behavioral heat defenses including vasodilation to promote heat loss, avoidance, suppression of BAT thermogenesis, and postal extension to promote heat loss.

that Pdyn+ and Penk+ populations overlap, as the sum of the Fos+ cells that are Pdyn+ and Penk+ exceeds 100%. Fos labeled, warm-activated, neurons are a subset of total Pdyn+ and Penk+ PBN neurons. These findings are consistent with previous reports that implicated glutamatergic FoxP2+ and Pdyn+ neurons in the dorsal lateral PBN in responding to warmth (*Geerling et al., 2016*).

VGLUT2+, Pdyn+, and Penk+ neurons from the PBN project to POA including the VMPO (*Figure 3* and *Figure 2—figure supplement 1K*). Pdyn+$^{PBN \to POA}$ neurons represent a subset of the VGLUT2+$^{PBN \to POA}$ population. Results obtained with retro-AAV Cre-switch, red (tdTomato) to green (GFP), injections into the POA revealed that all of the projecting PBN neurons are VGLUT2+ and a subset are Pdyn+ (*Figure 2—figure supplement 1A–D*). Gi DREADD mediated inhibition of PBN VGLUT2+ neurons but not of Pdyn+ PBN neurons (*Figure 5* and *Figure 4—figure supplement 1*) blocks vasodilation is response to thermal challenge. Further, results examining the expression of *Pdyn* and *Penk* in neurons retrogradely labeled with GFP from the POA (*Figure 2—figure supplement 2*) showed partially overlapping expression of GFP with both *Pdyn* and *Penk*. GFP+ cells that not labeled by either *Pdyn* or *Penk* were also evident. Taken together these results indicated that Pdyn+ and Penk+ PBN→POA neurons represent subsets of the glutamatergic warm-activated PBN neurons. A recent report presented data indicating the Pdyn vs Cck expression in the PBN marked a functional division of neurons driving vasodilation and BAT regulation (*Yang et al., 2020*). In contrast to those results, we found that *Pdyn* and *Cck* expression overlap in many lateral PBN neurons and that activation of Pdyn+$^{PBN \to POA}$ neurons induces tail vasodilation. To examine Cck and Pdyn expressing PBN populations, Yang et al. combined IHC for DynA with AAV based recombines reporter in Cck-Cre mice as a proxy expression of Cck. Based on this hybrid approach they reported minimal overlap of Dyn immunoreactivity and reporter expression. In contrast, we used FISH for mRNA for both peptides and found 70% overlap of *Pdyn* and *Cck* labeled LPBN neurons. The discordance of the two observation may rest in part with differing techniques (FISH vs IHC) used to examine peptide expression, the use of the Cck-Cre mouse line, underlying biological factors, such as variable peptide expression levels under different conditions, and known challenges in IHC staining vs mRNA labeling for cell identification. Future studies will need to employ RNAseq or other high resolution genetic methods to more clearly define PBN-POA cell identities, as is largely now accepted as a more thorough way, together with in situ to cluster and quantify neuronal groups within a given brain region.

The PBN projects to multiple areas in the POA and regulates other homeostatic processes including water balance and arousal (*Gizowski and Bourque, 2018*; *Qiu et al., 2016*). The studies presented here show PBN projections to the VMPO, which contains warm-activated neurons involved in regulating body temperature (*Tan et al., 2016*). Although we implanted fiberoptics above the VMPO light likely reach immediately adjacent regions of the POA, such as the MnPO, and PBN→POA projections were also observed in these areas. PBN to POA projections may be important in an array of homeostatic process. Recently, a connection of warm-activated neurons in the POA to promotion of sleep state has been described, and a role for temperature information from the PBN to the POA in promoting sleep has been suggested (*Harding et al., 2020*; *Harding et al., 2018*). Also, activation of neurons in the ventral lateral preoptic can also induce sleep and hypothermia (*Kroeger et al., 2018*). A potential role of PBN→POA projections in promoting sleep would suggest a bidirectional relationship of sleep by the PBN because PBN neurons are also able to cause arousal. A key to resolving the many roles of PBN neurons may lie in further understanding potential anatomical segregation of functionally discrete PBN circuit pathways. Anatomic segregation of pathways has been described for thermal information conveyed by the PBN, with cold responsive PBN separated from warm responsive neurons, which are located relatively caudal in the PBN (*Geerling et al., 2016*).

Likely postsynaptic targets in POA for warm-activated PBN cells include the recently identified warmth activated neurons in the POA that express the neuropeptides brain-derived neurotrophic factor (BDNF) and pituitary adenylate cyclase-activating polypeptide (PACAP) (*Tan et al., 2016*) as well as neurons in nearby areas implicated in mediating various homeostatic functions. Yang et al. show that blocking glutamatergic neurons in the POA blocked the effects on body temperature of activating PBN→POA (*Yang et al., 2020*). Also, prior studies have shown that hM3-Gq-DREADD induced stimulation of glutamatergic VMPO neurons (expressing the receptor for the hormone leptin) causes a reduction in core body temperature similar in magnitude to the effect seen by activation of Pdyn$^{PBN \to POA}$ terminals we observed in the present study. Further, activation of leptin

receptor expressing VMPO neurons also causes mice to display similar postural extension behavior as we observed following activation of Pdyn$^{PBN\to POA}$ terminals (*Figure 7*; *Yu et al., 2016*). Glutamatergic neurons in MnPO can drive vasodilation and may also be targets of the PBN warm-activated cells (*Abbott and Saper, 2018*).

Experiments presented here indicate that opioid signaling is not required for the rapid change in body temperature in response to activation of PBN neurons; however, evidence suggests that opioid systems may have important roles in regulation of body temperature and metabolism. Injections of opioid receptor agonists into the POA have been shown to alter body temperature indicating that opioid receptors, either pre- or post-synaptic to the PBN terminals, may have important functional neuromodulatory roles in thermoregulation (*Xin et al., 1997*). Further, POA KOR signaling was found to modulate body temperature and weight loss during calorie restriction (*Cintron-Colon et al., 2019*). Further supporting roles for opioid signaling in linking body temperature and metabolism, deletion of the KOR gene alters weight gain induced by a high fat diet by modulating metabolism (*Czyzyk et al., 2010*).

## PBN to POA projections regulate body temperature by evoking physiological and behavioral responses

Here we report results obtained in awake freely behaving mice that demonstrate the role of PBN to POA projecting neurons in driving physiological and behavioral responses to warm thermal challenge. Selective photostimulation of PBN→POA terminals in the three Cre lines (Pdyn, Penk, and VGLUT2) caused a robust and rapid decrease in body temperature (*Figure 4*). Thermal imaging paired with photoactivation of terminals revealed that the decrease in body temperature was due to heat loss via rapid vasodilation and suppression of BAT thermogenesis (*Figure 4G–L*). We found that hM4-Gi-DREADD mediated inhibition of VGLUT2+ (*Figure 5*), which encompasses both the Penk and Pdyn positive cells, but not Pdyn+ PBN neurons (*Figure 5—figure supplement 1*) alone blocked vasodilation in response to warm thermal challenge in awake animals. Activation of Penk+ or Pdyn+ PBN→POA terminals leads to rapid vasodilation and hypothermia (*Figure 4* and *Figure 4—figure supplement 1*) indicating that a subset of the VGLUT2+$^{PBN\to POA}$ population is sufficient to mediate vasodilation and suppress BAT activation. Taken together, our results demonstrate the necessity and sufficiency of transmission from VGLUT2+ PBN neurons to the POA for physiological responses to thermal heat challenge. Dyn peptide expression as marker of a separation of PBN neurons regulating BAT form those regulating heat loss by vasodilation as suggested by Yang et al. was not supported by the results in our experiments. In the presented report we did not examine the functional roles of Cck expressing neurons but did examine Penk+ PBN neurons. We found differing magnitudes of responses to activation of Pdyn+ or Penk+ terminals in the POA rather than categorical differences in the responses for the parameters examined. The results reported by Yang et al. are, overall, highly consistent with the results we present here, and there is divergence on Pdyn as a marker of functional separation in thermal defense circuits in the PBN. Future studies may help resolve if heat defense circuitry bifurcates at the level of the PBN using more genetically defined cell types, or through downstream activity in neurons in the POA mediated via specific neurotransmitters.

In rodents, thermal heat stress evokes behavioral changes including grooming, suppression of physical activity, postural changes (postural extension), and thermal seeking (*Roberts, 1988*). We found that activation of Pdyn$^{PBN\to POA}$ terminals can mediate these behaviors, including markedly suppressed locomotor activity and postural extension (*Figure 7*). Lesions of POA have been shown to block postural extension in response to warmth (*Roberts and Martin, 1977*), and selective activation of subpopulations of POA neurons evokes postural extension behavior (*Yu et al., 2016*). Many of the behavioral defenses appear to be due to activation of cells in the POA by PBN terminals.

## Activation of PBN to POA projecting neurons drives avoidance but does not promote thermal cool seeking

The PBN and the POA have been found to play important roles in thermal seeking behaviors, but the neural circuitry involved remains poorly understood. Warmth activated neurons within the POA have previously been found to drive a temperature preference *Tan et al., 2016*; however, the role of the POA in driving thermal seeking behaviors remains unclear. In contrast, prior studies using lesion

approaches in the POA did not block thermal seeking behaviors (*Almeida et al., 2006*; *Almeida et al., 2015*; *Matsuzaki et al., 2015*). Studies examining the role of PBN in other aversive stimuli have found roles for the PBN in encoding valence and engaging motivational systems to drive avoidance without disruption of behaviors driven by sensory input. For example, functional silencing of LPBN Calcitonin gene-related peptide expressing neurons suppressed pain escape behavior; however, sensory reflex responses (paw withdrawal latency) remained intact (*Han et al., 2015*). In this example, disruption of PBN circuit activity blocks the expression of avoidance behaviors but not the transmission of sensory input.

The PBN may play a similar role, driving avoidance/escape behavior without altering sensation, in thermal defense. Muscimol mediated inhibition of PBN blocks temperature preference behavior, and a spinothalamic pathway independently conveys temperature information (*Yahiro et al., 2017*). We found that stimulation of PBN→POA terminals engages affective and motivational circuitry driving avoidance (*Figure 6*). Photoactivation of Pdyn^PBN→POA terminals did not, however, induce a change in thermal preference for cooler temperatures (*Figure 7F,G*). In the context of previous studies, we interpret this to suggest that the coolness of the arena (20°C) as transduced by sensory pathways remains aversive despite the decrease in body temperatures evoked by the same photostimulation. Taken together with the literature, the results presented here support the conclusion that PBN neurons are necessary, but activation of this pathway (PBN to POA) alone is not sufficient for expression of cold seeking behaviors. Thermal seeking may also require information from additional neural circuits, with the PBN encoding valence. An alternative is that additional targets of PBN neurons outside the POA may be required to engage thermal cool seeking behaviors, and those targets were not affected by our experimental photostimulation of POA terminal fields. Supporting the notion that areas outside of the POA are required for thermal seeking, animals with POA lesion display amplification of motivated behaviors relating to thermal regulation due to impaired ability to defend core body temperature, and thus dependence on ambient temperature (*Lipton, 1968*; *Satinoff et al., 1976*). Future efforts will be necessary to understand the roles of POA and PBN circuits in modulating thermal motivated behaviors more fully.

## The endogenous opioid system is not required for acute effects of PBN neuron activation on body temperature

Opioid receptor modulation by agonists and antagonists has effects on body temperature regulation, acting at both central and peripheral sites through mu, kappa, and delta receptors (*Baker and Meert, 2002*). Specific effects of centrally administered mu and kappa antagonists on body temperature suggested a tonic balance between mu and kappa systems in maintaining body temperature (*Chen et al., 2005*). Here we examined the potential roles of the endogenous opioid system in the acute hypothermic response evoked by activation of PBN→POA terminals in the POA by blocking opioid signaling with naloxone, naltrexone, or norBNI (*Figure 4—figure supplement 1*). None of the selective opioid antagonists we used here significantly altered the response to acute stimulation of PBN terminals in the POA. One explanation for this lack of effect is that the PBN neuronal populations we examined are glutamatergic, and glutamate is known to be a key neurotransmitter for thermal regulation in the POA (*Nakamura and Morrison, 2010*). A role for the opioid system may be evoked by sustained changes in environmental temperature and may play a role in maintaining thermal set point in a modulatory capacity or play roles in context of altered metabolism or sleep. Additionally, our photo-activation paradigm might not be sufficient to produce endogenous opioid peptide release in these neurons. This is unlikely, however, given that our recent efforts in another region have shown that comparable photostimulation was sufficient to evoke both endogenous dynorphin and enkephalin release in vivo (*Al-Hasani et al., 2018*). Future studies with additional approaches and more sensitive peptide sensors may reveal further insights regarding the role of endogenous opioids in this circuitry.

## Conclusions and future directions

Previous studies have found that prior application of opioid receptor agonists affects the response of body temperature to opioid antagonists (*Baker and Meert, 2002*) and that environmental temperature, warm or cold, can dramatically alter the responses to centrally administered opioid peptides (*Handler et al., 1994*). Here we identified a potential source for multiple opioid peptides in the

thermoregulatory neurocircuitry and delineated a role for the neurons expressing Pdyn and Penk in regulating body temperature. How these neuromodulators are involved in regulating body temperature and the target neurons will require further experimentation to delineate. How opioidergic circuits and signaling contribute to processes involving thermal regulation and dysregulations, such as during opiate withdrawal and alterations in calorie intake, merit further study. In sum, we report here that Pdyn+, Penk+, and VGLUT2+ PBN neurons project to the POA, mediate physiological (vasodilation, suppression of thermogenesis) thermal defenses, drive behavioral thermal response behaviors (suppression of locomotion, postural changes), and drive aversion. The presented results will enable further studies to understand how homeostatic thermal regulation interacts with the motivational circuitry to drive behavior, provide targets for experiments testing the roles of neuromodulation of thermosensory pathways to regulate energy expenditure in balance with environmental factors, and help inform our understanding of how organisms balance competing interests, such as food intake, physical activity, and environmental conditions when selecting behaviors.

An area of future investigation will be to examine if subpopulations of POA cells may independently drive individual behavioral and physiological components of thermal regulation such as suppression of locomotion, postural extension, vasodilation, and alterations in metabolism. Yang et al. report that functional division of the circuits mediating aspects of physiological responses to warmth defense at the level of the PBN (*Yang et al., 2020*).

Although a functional division in the PBN is plausible, our results (particularly based on Pdyn expression) suggest that this conclusion warrants further study using high resolution approaches. It is also important to note that we recently observed that modulation of Pdyn PBN neurons can replicate behavioral effects that were once attributed solely to CGRP neurons in the PBN, and while these two populations are genetically distinct, similar behavioral effects were observed (*Bhatti et al., 2020*). Interestingly, Yang et al. observe hyperthermia in response activation of VGLUT2+ PBN neurons in some animals but with activation VGLUT2 PBN→POA projecting cells raising questions for future study about hypothermia activated PBN neurons and what their projection targets are. Taken together, the results we present demonstrate that PBN neurons, expressing VGLUT2 also express Penk, and/or Pdyn, project to the POA, and drive behavioral and physiological thermal heat coping behaviors.

# Materials and methods

## Key resources table

| Reagent type (species) or resource | Designation | Source or reference | Identifiers | Additional information |
|---|---|---|---|---|
| Antibody | Alexa Fluor 633 goat polyclonal anti-rabbit IgG | Invitrogen | Cat# A-21070, RRID:AB_2535731 | 1:1000 |
| Antibody | Phospho-c-Fos (Ser32) Rabbit monoclonal | Cell Signaling Technology | Cat# 5348S RRID:AB_10557109 | 1:500 |
| Chemical compound, drug | Clozapine N-oxide dihydrochloride | Hellobio | Cat# HB6149 | |
| Chemical compound, drug | Naloxone hydrochloride | Tocris | Cat# 0599 | |
| Chemical compound, drug | Naltrexone hydrochloride | Sigma-Aldrich | Cat# N3136 | |
| Chemical compound, drug | Norbinaltorphimine (norBN) | Sigma-Aldrich | Cat# N1771 | |
| Chemical compound, drug | U50,488 (U50) | Sigma-Aldrich | Cat# D8040 | |
| Strain, strain background (*Mus musculus*) | *Pdyn^{tm1.1(Cre)Mjkr}* | Gift from Brad Lowell, Harvard | RRID:MGI:5562890 | |

*Continued on next page*

*Continued*

| Reagent type (species) or resource | Designation | Source or reference | Identifiers | Additional information |
|---|---|---|---|---|
| Strain, strain background (*Mus musculus*) | B6;129S-*Penk*^tm2(Cre)Hze^/J | The Jackson Laboratory | RRID: IMSR_JAX: 025112 | |
| Strain, strain background (*Mus musculus*) | *Slc17a6*^tm2(cre)Lowl^/J | Gift from Brad Lowell, Harvard | RRID: IMSR_JAX: 028863 | |
| Strain, strain background (*Mus musculus*) | B6.Cg-*Gt(ROSA) 26Sor*^tm14(CAG-tdTomato)Hze^/J | The Jackson Laboratory | RRID: IMSR_JAX: 007914 | |
| Strain, strain background (*Mus musculus*) | C57BL/6J | The Jackson Laboratory | RRID: IMSR_JAX:000664 | |
| Strain, strain background (*AAV5*) | AAV5-EF1a-DIO-hChR2(H134R)-EYFP | Washington University Hope Center Viral Vector Core | N/A | $(2.5 \times 10^{13}$ vg/ml) |
| Strain, strain background (*AAV5*) | AAV5-hSyn-DIO-hM4D (Gi)-mCherry | Addgene | Addgene_44362-AAV5 | $(7 \times 10^{12}$ vg/ml) |
| Strain, strain background (*rAAV2-retro*) | AAV2-retro-DIO-ChR2-eYFP | Washington University Hope Center Viral Vector Core | N/A | $(2.8 \times 10^{12}$ vg/ml) |
| Strain, strain background (*AAV*) | AAV-retro-CAG-FLEX-tdTomato-WPRE | Addgene | Addgene_51503-AAVrg | $(1 \times 10^{13}$ vg/ml) |
| Strain, strain background (*AAV2*) | AAV2-retro-DO_DIO-tdTomato_EGFP-WPRE-pA | Addgene | RRID:Addgene_37120 | $(8 \times 1012$ GC/ml) |
| Strain, strain background (*AAV5*) | AAV5-Ef1a-DIO-eYFP | Washington University Hope Center Viral Vector Core | N/A | $(1.4 \times 10^{13}$ vg/ml) |
| Strain, strain background (*rAAV2-retro*) | AAV2-retro-GFP-Cre | Washington University Hope Center Viral Vector Core | N/A | $(3 \times 10^{13}$ vg/ml) |
| Strain, strain background (*AAV5*) | AAV5/hSyn-dio-hm4D(Gi)-mcherry | Addgene | RRID:Addgene_44362 | $(7.8 \times 1012$ vg/ml) |
| Sequence-based reagent | RNAscope probe *Pdyn* | Advanced Cell Diagnostics | accession number NM_018863.3 | probe region 33–700 |
| Sequence-based reagent | RNAscope probe *Penk* | Advanced Cell Diagnostics | accession number NM_001002927.2 | probe region 106–1332 |
| Sequence-based reagent | RNAscope probe *Slc17a6* | Advanced Cell Diagnostics | accession number NM_080853.3 | probe region 1986–2998 |
| Sequence-based reagent | RNAscope probe GFP | Advanced Cell Diagnostics | accession number AF275953.1 | probe region 12–686 |
| Sequence-based reagent | RNAscope probe Cre | Advanced Cell Diagnostics | accession number KC845567.1 | probe region 1058–2032 |
| Sequence-based reagent | RNAscope probe *Cck* | Advanced Cell Diagnostics | accession number NM_031161.3 | probe region 23–679 |
| Software, algorithm | ImageJ | NIH | RRID: SCR_003070 | |
| Software, algorithm | ResearchIR | FLIR Systems Inc | N/A | |
| Software, algorithm | Leica Application Suite Advanced Fluorescence | Leica Microsystems | N/A | |
| Software, algorithm | Bonsai | Bonsai-rx.org | N/A | |
| Software, algorithm | DeepLabCut | Mathis Lab (*Mathis et al., 2018*) | N/A | |

## Contact for reagent and resource sharing

Further information regarding reagents and resources may be directed to Aaron Norris, norrisa@-wustl.edu, or Michael Bruchas, mbruchas@uw.edu.

## Experimental model and subject details

Adult (25–35 g, older than 8 weeks of age during experiments) male and female Pdyn-Cre (RRID: MGI:5562890) (*Krashes et al., 2014*), Penk-Cre (RRID: IMSR_JAX: 025112) (*Harris et al., 2014*), Ai14-tdTomato (*Madisen et al., 2010*), and VGLUT2-Cre (*Vong et al., 2011*) mice (species *Mus musculus*) were group housed (no more than five littermates per cage) and allowed food and water ad libitum. Mice were maintained on a 12 hr:12 hr light:dark cycle (lights on at 7:00 am). All procedures were approved by the Animal Care and Use Committee of Washington University and adhered to NIH guidelines. The mice were bred at Washington University in Saint Louis by crossing the Pdyn-Cre, Penk-Cre, Ai14-tdTomato, and VGLUT2-Cre mice with C57BL/6 (RRID: IMSR_JAX:000664) wild-type mice and backcrossed for seven generations. Additionally, where needed, Pdyn-Cre and Penk-Cre mice were then crossed to Ai14-tdTomato mice on C57BL/6 background. Male and female mice were included and analyzed together.

## Stereotaxic surgery

Mice were anesthetized in an induction chamber (4% isoflurane), placed in a stereotaxic frame (Kopf Instruments), and anesthesia was maintained with 2% isoflurane. Mice were then injected bilaterally using a blunt needle Neuros Syringe (65457–01, Hamilton Com.) and syringe pump (World Precision Instruments) according to the injection schemes in the table below. The animal was kept in a warmed recovery chamber until recovery from anesthesia before being returned to its home cage.

| Virus | Virus volume | Brain region/coordinates |
| --- | --- | --- |
| AAV5-EF1a-DIO-hChR2(H134R)-EYFP (Hope Center Viral Vector Core, viral titer $2.5 \times 10^{13}$ vg/ml) | 150 nl | PBN, bilateral, (AP −5.00, ML ± 1.35, DV −3.50) |
| AAV5-hSyn-DIO-hM4D(Gi)-mCherry (Addgene, viral titer $7 \times 10^{12}$ vg/ml) | 150 nl | PBN, bilateral, (AP −5.00, ML +1.35, DV −3.50) |
| AAV2-retro-DIO-ChR2-eYFP (Hope Center Viral Vector Core, viral titer $2.8 \times 10^{12}$ vg/ml) | 100 nl | POA, unilateral, (+0.45 AP, +0.25 ML, −4.90 DV) |
| AAV2-retro-CAG-FLEX-tdTomato-WPRE (Addgene, viral titer $1 \times 10^{13}$ vg/ml) | 100 nl | POA, unilateral, (+0.45 AP, +0.25 ML, −4.90 DV) |
| AAV5-EF1a-DIO-eYFP (Hope Center Viral Vector Core, viral titer $1.4 \times 10^{13}$ vg/ml) | 150 nl | PBN, bilateral, (AP −5.00, ML +1.35, DV −3.50) |
| AAV2-retro-GFP-Cre (Hope Center Viral Vector Core, viral titer $3 \times 10^{13}$ vg/ml) | 100 nl | POA, unilateral, (+0.45 AP, +0.25 ML, −4.90 DV) |
| AAV-retro-DO_DIO-tdTomato_EGFP-WPRE-pA (Addgene, viral titer $8 \times 10^{12}$ GC/ml) | 100 nl | POA, unilateral, (+0.45 AP, +0.25 ML, −4.90 DV) |
| AV5/hSyn-dio-hm4D(Gi)-mcherry ($7.8 \times 10^{12}$ vg/ml) | 150 nl | PBN, bilateral, (AP −5.00, ML ± 1.35, DV −3.50) |

150 nl injections were injected at a rate of 30 nl/min, while 100 nl injections were injected at a rate of 20 nl/min. The injection needle was withdrawn 5 min after the end of the infusion. For anatomic experiments, mice that received unilateral or bilateral injections did not undergo further surgical procedures. For all behavioral experiments, mice underwent bilateral injections, implantations of a fiber optic for photostimulation over POA, and were implanted with a wireless IPTT-300 temperature transponder (Bio Medic Data Systems) subdermally directly rostral to right hindleg.

For photostimulation of PBN to POA projections, mice were injected with AAV5-EF1a-DIO-hChR2 (H134R)-EYFP and were allowed 6 weeks for sufficient proteins to reach distal axons. Mice were then implanted with mono fiber optic cannulas (ChR2 mice: Thor Labs, 1.25 mm OD ceramic ferrule, 5 mm cannula with 200 μm OD, 0.22 NA) in the VMPO (+0.45 AP, +0.25 ML, and −4.60 DV for ChR2 mice). The fiber optic implants were affixed using Metabond (Parkell). Mice were allowed 7 days of recovery before the start of behavioral experiments. Viral injection coverage and optical fiber

placements were confirmed in all animals using fluorescent microscopy in coronal sections (30 µm) to examine injection and implantation sites. Data from mice with incomplete viral coverage (i.e. unilateral expression of ChR2-eYFP in the PBN) or inaccurate optical fiber placement were excluded. Data from mice with bilateral PBN viral coverage and optical fiber placements near midline position over the POA were included in the study.

## Anatomical tracing

For anterograde viral tracing experiments, virus (AAV5-EF1a-DIO-hChR2(H134R)-EYFP or AAV5-EF1a-DIO-eYFP were used in our experiments) was injected at least 6 weeks prior to transcardial perfusions with 4% paraformaldehyde to allow for anterograde transport of the fluorophore. For retrograde viral tracing experiments, after the virus (AAV2-retro-DIO-ChR2-eYFP, AAV2-retro-CAG-FLEX-tdTomato-WPRE, AAV2-retro-EF1a-DO_DIO-TdTomato_EGFP-WPRE-pA, or AAV2-retro-GFP-Cre) was injected, there was a 3-week wait prior to perfusion to allow sufficient time for retrograde transport of the virus.

## Warm temperature exposure

Ai14xPdyn-Cre and Ai14xPenk-Cre mice in the warm condition were placed in a clean cage wrapped by a circulating water blanket which was set to 38°C. Mice in the room temperature condition were placed in a clean cage in a 22–23°C room. Water was supplied ad libitum in all cages. Cages in the warm condition were given enough time to reach the target temperature as confirmed by a thermometer before mice were placed inside of them. Temperature exposures lasted for 4 hr, after which mice were immediately anesthetized with pentobarbital and transcardially perfused with 4% paraformaldehyde in phosphate buffer, and brains were subsequently collected.

## Immunohistochemistry

IHC was performed as previously described by *Al-Hasani et al., 2013*, *Kim et al., 2013*; *McCall et al., 2015*. In brief, mice were intracardially perfused with 4% PFA and then brains were sectioned (30 microns) and placed in 1× PB until immunostaining. Free-floating sections were washed in 1× PBS for 3 × 10 min intervals. Sections were then placed in blocking buffer (0.5% Triton X-100% and 5% natural goat serum in 1× PBS) for 1 hr at room temperature. After blocking buffer, sections were placed in primary antibody rabbit Phospho-c-Fos (Ser32) antibody (RRID:AB_10557109, 1:500 Cell Signaling Technology) overnight at room temperature. After 3 × 10 min 1× PBS washes, sections were incubated in secondary antibody goat anti-rabbit Alexa Fluor 633 (RRID: AB_2535731, 1:1000, Invitrogen) for 2 hr at room temperature, followed by subsequent washes (3 × 10 min in 1× PBS then 3 × 10 min 1× PB washes). After immunostaining, sections were mounted on Super Frost Plus slides (Fisher) and covered with Vectashield Hard set mounting medium with DAPI (RRID:AB_2336788, Vector Laboratories) and cover glass prior to being imaged on a Leica DM6 B microscope.

| Alexa fluor 633 anti-rabbit IgG | Goat | 1:1000 | Invitrogen | RRID:AB_2535731 |
|---|---|---|---|---|
| Phospho-c-Fos (Ser32) Rabbit mAb | Rabbit | 1:500 | Cell Signaling | RRID:AB_10557109 |

## Imaging and cell quantification

Brain sections in figures are labeled relative to bregma using landmarks and neuroanatomical nomenclature as described in 'The Mouse Brain in Stereotaxic Coordinates' (*Franklin and Paxinos, 2013*).

To quantify the number of cells expressing cFos, dynorphin, and/or enkephalin, cFos was labeled by Alexa Fluor 633, a fluorophore with emission in 610–800 nm (max 650 nm) range and preproenkephalin/prodynorphin were labeled by tdTomato with emission in the 540–700 nm (max 581 nm) range. All sections were imaged on a Leica DM6 B epifluorescent microscope using a Texas Red Filter Cube (Excitation: BP 560/40, Dichroic: LP 585, Emission: BP 630/75) for tdTomato visualization

and a CY5 Filter Cube (Excitation: BP 620/60, Dichroic: LP 660, Emission: BP 700/75) for Alexa Fluor 633. Images were obtained for each 30 µm section that contained neurons in the PBN.

We defined the boundaries of LPBN as follows. Sections between −5.0 and −5.4 rostral to bregma were imaged for LPBN exclusively. The superior cerebellar peduncle marked the medial and ventral boundaries of LPBN. The lateral boundary was marked by the ventral spinocerebellar tract, and the dorsal boundary was marked by the cuneiform nucleus.

All image groups were processed in parallel using ImageJ (RRID: SCR_003070, v1.50i) software. IHC was quantified as previously described (Al-Hasani et al., 2013; Kim et al., 2013). Briefly, channels were separated, an exclusive threshold was set, and positive staining for each channel was counted in a blind-to-treatment fashion using ImageJ. The counts from each channel were then overlaid and percent of co-labeled cells were reported.

### Fluorescent in situ hybridization (FISH)

Following rapid decapitation of mice, brains were flash frozen in −50°C 2-methylbutane and stored at −80°C for further processing. Coronal sections containing the PBN region, corresponding to the injection plane used in the behavioral experiments, were cut at 20 µM at −20°C and thaw-mounted onto Super Frost Plus slides (Fisher). Slides were stored at −80°C until further processing. FISH was performed according to the RNAScope 2.0 Fluorescent Multiple Kit User Manual for Fresh Frozen Tissue (Advanced Cell Diagnostics, Inc) as described by Wang, 2012 – see below. Slides containing the specified coronal brain sections were fixed in 4% paraformaldehyde, dehydrated, and pretreated with protease IV solution for 30 min. Sections were then incubated for target probes for mouse Pdyn (Pdyn, accession number NM_018863.3, probe region 33–700), Penk (Penk, accession number NM_001002927.2, probe region 106–1332), VGLUT2 (Slc17a6, accession number NM_080853.3, probe region 1986–2998), GFP (GFP, accession number AF275953.1, probe region 12–686), and/or Cre (Cre, accession number KC845567.1, probe region 1058–2032) for 2 hr. All target probes consisted of 20 ZZ oligonucleotides and were obtained from Advanced Cell Diagnostics. Following probe hybridization, sections underwent a series of probe signal amplification steps (AMP1–4) including a final incubation of fluorescently labeled probes (Alexa 488, Atto 550, Atto 647), designed to target the specified channel (C1–C3 depending on assay) associated with the probes. Slides were counterstained with DAPI and coverslips were mounted with Vectashield Hard Set mounting medium (Vector Laboratories). Alternatively, mice transcardially perfused with cold PBS and PFA with fixed brain tissue collected and sectioned at 30 µM as described previously were processed for FISH as above.

Images were obtained on a Leica DM6 B upright microscope (Leica), and Application Suite Advanced Fluorescence (LAS AF) and ImageJ software were used for analyses. To analyze images for quantification of Pdyn/Penk/VGLUT2 coexpression, each image was opened in ImageJ software, channels were separated, and an exclusive fluorescence threshold was set. We counted total pixels of the fluorescent signal within the radius of DAPI nuclear staining, assuming that each pixel represents a single molecule of RNA as per manufacturer guidelines (RNAscope). A positive cell consisted of an area within the radius of a DAPI nuclear staining that measured at least five total positive pixels. Positive staining for each channel was counted in a blind-to-condition fashion using ImageJ or natively in LAX software (Leica).

### Behavior

All behaviors were performed within a sound-attenuated room maintained at 23°C at least 1 week after the final surgery. For open field assays, lighting was stabilized at ~250 lux for aversion behaviors (Figures 6 and 7, and Figure 6—figure supplement 1A–C) and ~200 lux for body temperature change recordings and heat challenges (Figures 4 and 5, Figure 4—figure supplement 1, Figure 5—figure supplement 1, and Figure 6—figure supplement 1D,E). Movements were video recorded and analyzed using Ethovision XT 10 (Noldus Information Technologies). For all optogenetic experiments, a 473 nm laser (Shanghai Lasers) was used and set to a power of ~15 mW from the tip of the patch cable ferrule sleeve. All patch cables used had a core diameter of 200 µm and a numerical aperture of 0.22 (Doric Lenses Inc). At the end of each study, mice were perfused with 4% paraformaldehyde followed by anatomical analysis to confirm viral injection sites, optic fiber implant sites, and cell-type-specific expression.

### Real-time place aversion testing

We used four copies of a custom-made, unbiased, balanced two-compartment conditioning apparatus (52.5 × 25.5 × 25.5 cm) as described previously (*Jennings et al., 2013*; *Stamatakis and Stuber, 2012*). Mice were tethered to a patch cable and allowed to freely roam the entire apparatus for 30 min. Entry into one compartment triggered constant photostimulation at either 0 Hz (baseline trial), 2 Hz, 5 Hz, 10 Hz, or 20 Hz (473 nm, 10 ms pulse width) while the mouse remained in the light paired chamber. Entry into the other chamber ended the photostimulation. The side paired with photostimulation was counterbalanced across mice. Ordering was counterbalanced with respect to stimulation frequency and placement in each of four of the copies of behavior apparatus. Bedding in all copies of the behavior apparatus was replaced between every trial, and the floors and walls of the apparatus were wiped down with 70% ethanol. Time spent in each chamber and total distance traveled for the entire 30 min trial were measured using Ethovision 10 (Noldus Information Technologies).

### Core body temperature, vasodilation, and BAT thermogenesis recordings

We used transparent circular behavioral arenas (diameter = 14.5 inches, wall height = 21 cm) for experiments measuring core body temperature changes, vasodilation, and BAT thermogenesis suppression corresponding to optogenetic stimulation. Mice were tethered to a patch cable and allowed to habituate to the arena for 1 hr. Core body temperature measurements were made every 5 min beginning 5 min prior to turning the laser on. Laser frequencies of 2 Hz, 5 Hz, 10 Hz, and 15 Hz were used. Core body temperature measurements were made using a DAS-8007 Reader (Bio Medic Data Systems) which wirelessly read the temperature from a subdermally implanted IPTT-300 temperature transponder in each mouse (previously validated by *Langer and Fietz, 2014*).

Thermal imaging of mice was carried out using a FLIR E53 thermal imaging camera (FLIR Systems Inc) to record the 65 min trial. Fur over the intrascapular region was shaven to facilitate temperature readings of the interscapular BAT (*Crane et al., 2014*). Thermal imaging videos were scored in a blind-to-genotype/condition fashion using ResearchIR software (FLIR Systems Inc). Eye, tail, and BAT temperatures were read every minute for the first 35 min of each trial and every 5 min for the final 30 min. Tail temperature readings were taken ~1 mm away from the base of the tail. BAT temperature readings were taken at the warmest point of the intrascapular region. Eye temperature readings were taken at the warmest point of the eye. To quantify the postural extension during these experiments, an investigator reviewed each video and quantified in two-minute bins the percent time the mice were in an extended posture (sprawled on the bedding) and time in huddled position with their tail tucked under their bodies.

### Real-time place aversion testing

We used four copies of a custom-made, unbiased, balanced two-compartment conditioning apparatus (52.5 × 25.5 × 25.5 cm) as described previously (*McCall et al., 2015*; *Parker et al., 2019*). Mice were tethered to a patch cable and allowed to freely roam the entire apparatus for 30 min. Entry into one compartment triggered constant photostimulation at either 0 Hz (baseline trial), 2 Hz, 5 Hz, 10 Hz, or 20 Hz (473 nm, 10 ms pulse width) while the mouse remained in the light paired chamber. Entry into the other chamber ended the photostimulation. The side paired with photostimulation was counterbalanced across mice. Ordering was counterbalanced with respect to stimulation frequency and placement in each of four of the copies of behavior apparatus. Bedding in all copies of the behavior apparatus was replaced between every trial, and the floors and walls of the apparatus were wiped down with 70% ethanol. Time spent in each chamber and total distance traveled for the entire 30 min trial were measured using Ethovision 10 (Noldus Information Technologies).

### Locomotion changes with U50 and antagonists

Mice were habituated in a clear chamber (Cambro 18SFSCW135 CamSquare 18 Qt., Cambro City of Industry, Huntington Beach, CA, USA) and then placed on a plywood platform. Two LED lights were placed above the chamber to provide adequate lighting. Room temperature and lighting intensity remained consistent (22.4℃, 132 lux). Cameras (Camera body: ELP-USBFHD01M-SFV, 2.8–12 mm lens) recorded videos for 45 min at 60fps at a resolution of 1920 × 1080 p. Cameras were mounted

directly above the chamber and placed on a tripod perpendicular to the chamber. Video recording was controlled through a custom Bonsai Program (*Lopes et al., 2015*) to allow simultaneous video recording. Mice were allowed to roam the chamber freely upon being injected.

Mice locomotion was analyzed using DeepLabCut, a markerless pose estimator (*Mathis et al., 2018*). A mouse model was trained using Resnet-50 and k-means clustering on 75 frames from three videos. The model was trained to 250,000 iterations with an average test error of pixel error 5.71 pixels, a calculation of the distance between human labels versus labels predicted by DeepLabCut to determine the accuracy of the trained model. Videos were trained on a Dell workstation on Windows 10 Enterprise with 4.10 GHz Intel Xenon processor with 32 GB RAM and a NVIDIA Quadro RTX 5000 GPU.

A custom Python script (version 3.8) was created to quantify mice locomotion in terms of velocity in the form of pixels per second (NorrisLab 2020). Top-down videos were analyzed to determine cumulative moving averages for each condition; results were then averaged for 10 s. Results were compared to determine the effects of drug administration on locomotion to ensure appropriate CNS targeting.

## Drug administration

Clozapine N-oxide dihydrochloride (Hellobio) was made in sterilized distilled water and mice received an intraperitoneal (i.p) injection of water (vehicle) or CNO (2.5 mg/kg) and were placed in the heat challenge apparatus or thermal plate preference apparatus for 30 min of habituation prior to beginning the assay. Naloxone hydrochloride (Tocris) and naltrexone hydrochloride (Sigma-Aldrich) were dissolved in 0.9% saline. Penk-Cre and Pdyn-Cre mice received an i.p. injection of naloxone (5 mg/kg), naltrexone (3 mg/kg), or saline (vehicle) respectively and were placed back in their home cages for 30 min before being placed into behavioral arena. In experiments using norBN, norBNI dissolved in DMSO (10 mg/kg) was given IP approximately 24 hr prior to start of experiments and again 30 min immediately prior to start of the assay.

## Heat challenge

For chemogenetic inhibition experiments exposing mice to a heat challenge (*Figure 5* and *Figure 5—figure supplement 1*), we used a purpose-built, two-temperature water circulation apparatus to rapidly change the floor and wall temperatures of a square, transparent behavioral arena (15.25 × 15.25 × 19 cm). After drug or saline administration, mice were habituated to the arena at 20℃ for 30 min. The water flow to the arena was changed to 34℃, and the temperature of floor and walls rose quickly, reaching steady state in the first 4 min (time course of ambient temperature change can be seen in *Figure 5B*). The water flow to the arena was switched back to 20℃ after 15 min. Thermal imaging recording was obtained beginning 5 min prior to heat challenge and for 10 min post heat challenge for a total of 30 min. Thermal imaging videos were used to measure eye temperature, tail temperature, BAT temperature, and the temperature of the behavioral arena every minute throughout the 30 min heat challenge trial. Thermal imaging videos were scored in a blind-to-genotype/condition fashion.

## Thermal preference

For experiments presenting mice with a choice between two floor plate temperatures (*Figure 7F,G*), we used a purpose-built apparatus consisting of two fused Cold/Hot Plate Analgesia Meters (Columbus Instruments International) with plastic walls surrounding and dividing the plates to create two behavioral arenas with 4-inch width, 19.5-inch length, and 9-inch height of walls. One side of the behavioral arena was set to 26℃ and the other to 20℃. The side set to 20℃ was counterbalanced across mice. Pdyn-Cre mice were tethered to a patch cable and placed into a behavioral arena. Mice were allowed to roam the arena for 40 min before photostimulation. The laser frequency was set to 10 Hz and was left on for 20 min. Mice were kept in the behavioral arena for an additional 20 min post-stimulation. Time spent on each side for the entire 80 min trial was quantified using Ethovision 10.

## Open field test

For experiments quantifying distance moved upon photostimulation of Pdyn$^{PBN \to POA}$ (*Figure 7A–C*), we used a purpose-built 20in square behavior arena. Pdyn-Cre mice were tethered to a patch cable and placed into the behavioral arena. The laser frequency was set to 10 Hz and was left on for 20 min. Distance moved for the 20 min trial was quantified using Ethovision 10. Bedding in the arena was replaced between every trial, and the floors and walls of the arena were wiped down with 70% ethanol.

## Statistical analyses

All data are expressed as mean ± SEM. Statistical significance was taken as $^{*}p<0.05$, $^{**}p<0.01$, $^{***}p<0.001$, and $^{****}p<0.0001$ as determined by Student's t-test, one-way ANOVA, or a two-way repeated measures ANOVA followed by a Bonferroni post hoc tests as appropriate. Statistical analyses were performed in GraphPad Prism 7.0. For each experiment, control groups and statistics are described in the main text. All 'n' values represent the number of animals in a particular group for an experiment.

Experiments involving optogenetic stimulation of PBN inputs to POA using Pdyn-Cre, Penk-Cre, and VGLUT2-Cre mice (*Figures 4*, *6,* and *7*, *Figure 4—figure supplement 1*, and *Figure 6—figure supplement 1*) were replicated in three separate cohorts for each genotype. Chemogenetic inhibition experiments (*Figure 5* and *Figure 5—figure supplement 1*) were replicated in two separate cohorts of VGLUT2-Cre mice and two separate cohorts of wild-type mice. Warm temperature exposure with cFos immunohistochemical staining experiments (*Figure 1*) were performed in four separate iterations. Each iteration replicated the results of those prior to it, and data from each iteration was included in the overall statistical analysis of the experiment.

An investigator was blinded to allocation of groups in experiments whose data is shown in *Figure 1* (warm-induced cFos+ cell quantification), *Figure 1—figure supplement 1*/*Figure 2*/*Figure 2—figure supplement 2* (in situ hybridization quantification), and *Figure 4*/*Figure 4—figure supplement 1*/*Figure 5*/*Figure 5—figure supplement 1*/*Figure 7* (thermal video scoring).

## Acknowledgements

The authors thank Megan Votoupal for her technical assistance in mouse husbandry. This work was supported by a Foundation for Anesthesia Education and Research (FAER) Grant, and National Institute for Mental Health (NIMH) grant K08MH119538 and R21EY031269 to AJN, by R01MH11235505 and R37DA03339607 to MRB, P30DA048736, and by a Pilot Project Award from the Hope Center for Neurological Disorders at Washington University to AJN and MRB. The Mallinkrodt Foundation (MRB Professorship). The graphic summary illustration (*Figure 8*) was created by Percy Griffin with Astrid Rodriguez Velez in association with InPrint at Washington University School of Medicine.

## Additional information

### Funding

| Funder | Grant reference number | Author |
|---|---|---|
| National Institute of Mental Health | K08MH119538 | Aaron J Norris |
| National Institute of Mental Health | R37DA033396 | Michael R Bruchas |
| Hope Center for Neurological Disorders | | Aaron J Norris<br>Michael R Bruchas |
| National Eye Institute | R21EY031269 | Aaron J Norris |
| National Institute of Mental Health | | Michael R Bruchas |
| National Institute of Mental Health | R37DA03339607 | Michael R Bruchas |
| National Institute of Mental | P30DA048736 | Michael R Bruchas |

Health

The funders had no role in study design, data collection and interpretation, or the decision to submit the work for publication.

## Author contributions

Aaron J Norris, Conceptualization, Resources, Data curation, Software, Formal analysis, Supervision, Funding acquisition, Validation, Investigation, Visualization, Methodology, Writing - original draft, Project administration, Writing - review and editing; Jordan R Shaker, Conceptualization, Formal analysis, Investigation, Visualization, Writing - original draft, Writing - review and editing; Aaron L Cone, Investigation, Methodology; Imeh B Ndiokho, Investigation, Visualization; Michael R Bruchas, Conceptualization, Resources, Supervision, Funding acquisition, Writing - review and editing

## Author ORCIDs

Aaron J Norris https://orcid.org/0000-0001-7825-1756
Jordan R Shaker https://orcid.org/0000-0002-4496-3904
Aaron L Cone https://orcid.org/0000-0003-4411-6673
Imeh B Ndiokho https://orcid.org/0000-0003-1924-1368
Michael R Bruchas https://orcid.org/0000-0003-4713-7816

## Ethics

Animal experimentation: This study was performed in strict accordance with the recommendations in the Guide for the Care and Use of Laboratory Animals of the National Institutes of Health. All of the animals were handled according to approved institutional animal care and use committee (IACUC) protocols of Washington University (#19-0835).

## Decision letter and Author response

Decision letter https://doi.org/10.7554/eLife.60779.sa1
Author response https://doi.org/10.7554/eLife.60779.sa2

# Additional files

## Supplementary files

• Transparent reporting form

## Data availability

All data generated or analyzed during this study are included in the manuscript and supporting files.

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
