## [Decision Letter]

**Acceptance summary:**

Work by Norris et al. uses circuit-tracing and optogenetic and chemogenetic manipulations to examine the role of a parabrachial nucleus to preoptic area (POA) circuit in thermal defense behaviors. Specifically, the authors show that a population of dynorphin neurons (some also express enkephalin) that project to the POA and were previously shown to induce expression of c-Fos in response to warming (Geerling et al., 2016) drive a decrease in BAT and core body temperature and an increase in tail vasodilation as well as behavioral changes such as decreased locomotor activity and induction of extended body posture. The study fills in some of the important and still unresolved knowledge gaps in the identity of central circuits that direct thermal homeostasis.

**Decision letter after peer review:**

Thank you for submitting your article "Parabrachial Opioidergic Projections to Preoptic Hypothalamus Mediate Behavioral and Physiological Thermal Defenses" for consideration by *eLife*. Your article has been reviewed by three peer reviewers, one of whom is a member of our Board of Reviewing Editors, and the evaluation has been overseen by Ronald Calabrese as the Senior Editor. The following individuals involved in review of your submission have agreed to reveal their identity: Jan Siemens (Reviewer #2); William Wisden (Reviewer #3).

The reviewers have discussed the reviews with one another and the Reviewing Editor has drafted this decision to help you prepare a revised submission.

Summary:

Work by Norris et al. use circuit-tracing and optogenetic and chemogenetic manipulations to examine the role of a parabrachial nucleus (PBN) to preoptic area (POA) circuit in thermal defense behaviors. Specifically, the authors show that a population of dynorphin neurons (some also express enkephalin) that project to the POA and were previously shown to induce expression of c-Fos in response to warming (Geerling et al., 2016) drive a decrease in BAT and core body temperature and an increase in tail vasodilation as well as behavioral changes such as decreased locomotor activity and induction of extended body posture. The study fills in some of the important and still unresolved knowledge gaps in the identity of central circuits that direct thermal homeostasis.

Essential revisions:

1) This PBN-POA circuit is also the subject of a similar study by Yang et al., currently released on bioRxiv (https://doi.org/10.1101/2020.06.11.138370). The studies are complementary and reinforcing, but also show differences that the authors should discuss including the role of PBN to POA neuronal populations in tail vasodilation. This function is assigned by Yang et al. to a population of PBN neurons that instead express cholecystokinin.

2) With respect to defining the PBN aspect of the circuit, the authors should report more precisely the scope of the c-Fos expression within the dynorphin, enkephalin and VGLUT2 populations. For example, they should express the c-Fos positive population as a fraction of Dyn- and Enk-positive PBN populations. This would reveal the possibility that some dynorphin neurons in this region may project to other areas and have other functions. The authors should mention/discuss the potential heterogeneity of this population. Related to this, are all of the EnK^+^ neurons that project to the POA Dyn+? A Venn diagram could help clarify the relationship of the four markers (Dyn, VGLUT2, c-Fos, Enk). Finally, the authors should include whether Dyn or Enk expression in PBN changes over development. Concerns relate to whether tomato expression differs from Dyn, Enk, or actual Cre expression in the adult.

3) With respect to defining the POA aspect of the circuit, the authors are focused on VMPO, but based on the data presented, it appears that other nearby regions may also have been targeted in their experiments (Figure 3C)? If so, what happens to body temperature when optogenetically activating those extra-VMPO Dyn-fiber terminals? MnPO/AVPe and OVLT have been implicated in osmo- and thirst regulation and may modulate heart rate (as described in cited papers and reviews), which could affect body temperature. Did the authors measure heart rate change upon optogenetic stimulation of PBN (Dyn/Enk) fibers terminating in the VMPO?

4) Data demonstrating the requirement for the circuit is weakened by an apparent lack of specificity in chemogenetically inhibiting the VGLUT2 population. Do VGLUT2^+^ neurons in this area project elsewhere? Specific inhibition of the Dyn+ neuron-POA circuit would be more appropriate.

5) The authors frequently mention a role for opioids in the POA in thermoregulation and perform an experiment to test whether dynorphin and enkephalin in this particular PBN-POA circuit has a role. It may be the case that the peptides released by these projections do not have a role, but the authors need to demonstrate positive controls for the antagonists. Are they able to cross the blood brain barrier and was there sufficient time for the drugs to enter the brain before optogenetic stimulation? The authors could also consider trying other antagonists particularly for KOR such as DIPPA (Cintron-Colon et al., Current Biology, 2019).

[Editors' note: further revisions were suggested prior to acceptance, as described below.]

Thank you for resubmitting your work entitled "Parabrachial Opioidergic Projections to Preoptic Hypothalamus Mediate Behavioral and Physiological Thermal Defenses" for further consideration by *eLife*. Your revised article has been evaluated by Ronald Calabrese (Senior Editor) and a Reviewing Editor.

The manuscript has been improved but there are some remaining issues that need to be addressed before acceptance, as outlined below:

The reviewers appreciate the importance of this work to understanding the neural circuitry of thermal defense behaviors and some of the revisions that were made by the authors did help clarify the manuscript, as outlined in attached reviews. On the other hand, a couple of the major issues raised by the reviewers were not adequately addressed (#1 and #2 in previous review). 1) The authors have advanced rather than resolved a discrepancy between their new data and that of Yang et al. (Science Advances) regarding Cck and Pdyn expression, with the authors claiming considerable overlap and Yang et al. claiming distinct populations. It is important to clarify this point and also with respect to the physiological mechanisms of thermal defense behavior. As it stands now the authors have presented a contradiction. 2) The percentage of Pdyn+ neurons that expressed c-Fos with warming and the percentage of Pdyn+ neurons that project to POA that expressed c-Fos with warming should be reported. The same should be reported for PenK^+^ neurons. Results from this analysis could alter the representations shown in Figure 8. 3) It should be noted that the new data in Figure 2—figure supplement 2 in theory provide important new information, but the specific signals within the images are not robust making interpretations of them difficult. i.e. There is one GFP cell clearly visible in panel B and what is background for panel E?

Reviewer #1:

The study by Norris et al. uses optogenetic /chemogenetic manipulations with physiological measurements and behavior as well as circuit tracing to demonstrate that excitatory neurons in the LPBN, some of which express Pdyn and Penk (partially overlapping), and that project to the POA are important for heat defense behaviors.

1) Explain how their work fits with work by Yang et al., which is now published in Science Advances.

Yang et al. claim that CcK^+^ and Pdyn+ neurons are distinct populations. The new in situ in Figure 2—figure supplement 2 presumably indicates 70% of Pdyn+ cells are CcK^+^, but the image shown is not entirely convincing. Whether the signal is over background in the area where Pdyn+ cells are located is unclear even based on the inset. Nevertheless, the finding does not entirely reconcile the differences with the Yang et al. study. Yang et al. claim that Pdyn+ cells are involved in a defensive heat response via an inhibition of iBAT while CCK neurons are involved in a defensive heat response involving vasodilation. Activation of Pdyn neurons does not evoke tail vasodilation while activation of Cck cells does. Yang et al., also show that TeNT inhibition of Pdyn neurons affects core temp and iBAT temp and very modestly tail temp, while inhibition of Cck neurons had effects on core temp and fever, but not tail temp. This suggests Pdyn are required for IBAT but Cck cells are involved but not required for tail vasodilation. Norris et al. show here that activation of the Pdyn and Penk to POA projections cause changes in core, BAT and tail temp. They examine only tail temp with inhibition of Pdyn or Penk neurons, which they show does not change in either case. In sum, the two studies differ in whether the Pdyn neurons are sufficient to evoke tail temp change and it is unclear how the Yang et al. studied a population of Pdyn neurons distinct from Cck while Norris et al. studied Pdyn neurons that largely express Cck.

2) Illustrate better the overlap of c-Fos within the three populations, whether all PenK^+^ neurons that project to POA are also Pdyn+ and whether Pdyn or Penk expression from the tomato mouse cross differs from adult in situ or adult virus injection (developmental change).

In Figure 1, the authors cross Ai14 tomato mouse to the Cre lines and look at c-Fos with warmth exposure. However, in the Figure 1—figure supplement 1 they do not address whether Ai14 tomato distribution differs from adult Penk or Pdyn rather they instead compare to an in situ probe for Cre in adult. This control is useful for comparing viral expression, but not for addressing overlap of c-Fos shown in Figure 1. To address development, it would been good to perform in situ for tomato with Penk/ Pdyn and c-fos in adult assuming they still have the animals.

In Figure 2—figure supplement 2 the GFP signal in panel B is surprisingly not very robust assuming this is representative. Very few neurons in the slice appear to be GFP+ and many are not where the arrows are clustered. It would be helpful if the reviewers showed on the image the areas that were used in the analyses. Was it only where the arrows were clustered?

Reviewer #2:

The authors have not adequately addressed the reviewers' comments and concerns. This is not to say that we, as reviewers, expect the authors to do all the experiments --or for that matter even change all the wording in the text-- as per reviewers' suggestion. But authors should at least address in the point-by-point rebuttal letter and state why they disagree with the suggestions and/or (i) did not need to or (ii) couldn't or (iii) did not want to implement the changes the reviewers suggested.

1) The authors suggest that Pdyn neurons in the PBN specifically relay temperature information to the POA/VMPO. If Pdyn is labeling neurons that are warm-activated, I would expect a substantial fraction of cfos-positive cells to overlap with pdyn. To assess this requires to calculate the cFos-positive population as a fraction of Dyn- and Enk-positive populations (e.g. what % of Dyn neurons is cfos positive upon warming) instead of expressing the cfos-fraction that is positive for dyn or enk (as has been done by the authors).

In the response the authors vaguely argue that this information can be found in Geerling et al., 2016. We looked it up and Geerling et al. suggest that around 20% of pdyn neurons are specifically induced to express cfos upon warming. Comparing the figure in Geerling et al., 2016 (Figure 7) with the equivalent one in this manuscript (Figure 1D) my best guess is that in the hands of the authors this fraction is even lower. And thus this fraction is misrepresented in the summary Figure 8: here in the cartoon it looks like the majority of warm activated cells are pdyn positive! This also renders the following sentence in the Discussion questionable "...A subset (Pdyn+ or PenK^+^) of the VGLUT2^+^ PBN→POA population is likely sufficient to mediate vasodilation and suppress BAT activation.." We disagree, in our view this means that the largest fraction of pdyn neurons is likely doing something else and not relaying temperature information. This possibly explains also the new data in Figure 5—figure supplement 1: blocking the pdyn population by Gi-DREADD does not have an effect (different to blocking vglut2-Cells).

Reversely, does this mean that their Gq-DREADD data (Figure 4I) is wrong and activating the pdyn population should also not activate tail vasodilation as suggested by Yang et al., biRxv or Science Advances 2020? Again, I like to reiterate that the authors don't need to do all what the reviewers suggest, but glossing over important points and not addressing them properly in the letter (and the manuscript) is not a good practice. This brings us to the next point:

2) The authors now do dual-color in situs to test what fraction of CCK neurons overlap with pdyn neurons to conclude that "..Findings from these experiments indicate 70% of Pdyn labeled neurons are also labeled by Cck probes. This congruency helps to resolve any discrepancy that a majority of these two neuronal populations are indeed overlapping…". We beg to differ: the authors Yang et al., 2020 use --as far as we can see-- the same pdyn-Cre mouse line as the authors and they don't see any tail vasodilation when activating these neurons (Figure 5B in the Science Advances publication) and this has nothing to do with any CCK expression. Again, this discrepancy may not be easily resolvable, but to hide it and to say CCK is a subset of these neurons and this resolves the discrepancy is not correct! This functional difference may --at least for some researchers- be important and thus should be spelled out.

3) The manuscript is very difficult to read with all the edits, "invalid citations" etc.

– The authors should read the manuscript carefully and correct before submission.

In summary, we still believe that the manuscript is valuable and adds to the Yang et al. study, but we feel the above aspects should be addressed, at least in the response letter, and the manuscript carefully edited before publication.

Reviewer #3:

It's been known for many years that warm receptors on the skin pass on information through the parabrachial relay glutamate neurons in the brainstem, and that these glutamate neurons project onwards to the preoptic hypothalamic area. The current paper is a functional anatomical study exploring in more detail the role of glutamate neurons in the parabrachial nucleus that project to the preoptic area to regulate defensive control of body temperature. The authors find that dynorphin and enkephalin release in the preoptic area are not needed, but glutamate, is for external warmth induced body cooling. Although not explicitly shown by the authors, this is probably one type of cell in the parabrachial that co-releases glutamate, dynorphin and enkephalin. The standard of work was high and detailed. A lot of technically excellent work is present tin the paper and it is beautifully presented.

The authors have adequately revised the manuscript and I have no further concerns

[Editors' note: further revisions were suggested prior to acceptance, as described below.]

Thank you for resubmitting your work entitled "Parabrachial Opioidergic Projections to Preoptic Hypothalamus Mediate Behavioral and Physiological Thermal Defenses" for further consideration by *eLife*. Your revised article has been evaluated by Ronald Calabrese (Senior Editor) and a Reviewing Editor.

The manuscript has been improved but there are some remaining issues that need to be addressed before acceptance, as outlined below:

1) The comparison of the Pdyn staining and the Pdyn-Cre line shown in Yang et al. Figure 2A indicates a high level of co-localization. Norris et al. have used the same Pdyn-Cre line in the present manuscript. Comparison of Pdyn and Cck in Yang et al. Figure 2L was done by staining the Cck-Cre mouse with the Pdyn antibody used in Figure 2A. The difference between Yang et al. and Norris et al. is therefore the difference between the Cck-Cre line (Yang et al) and the use of RNAScope (current manuscript). The authors should correct how they discuss this in the manuscript, as it now is stated as being due strictly to antibody staining by Yang et al. vs RNAScope used by the authors of the current manuscript.

2) As for showing the percentage of Pdyn and Penk neurons that express c-Fos based on data shown in Figure 1, an additional limitation (on top of those noted by the authors) to the interpretation of reporting cell counts as a percentage of either c-Fos or the marker would also be if the number of tomato+ cells observed using the Cre lines crossed to the reporter is not the same as the Pdyn or Penk population manipulated by the viruses. The authors addressed this concern in an earlier version in their response to the reviewers, but direct evidence in the paper is lacking. Nevertheless, providing the readers with some sense of the percentage of Pdyn or Penk neurons that were c-Fos positive from the images that they quantified in Figure 1 would be similarly useful.

---

## [Author Response]

Essential revisions:1) This PBN-POA circuit is also the subject of a similar study by Yang et al., currently released on bioRxiv (https://doi.org/10.1101/2020.06.11.138370). The studies are complementary and reinforcing, but also show differences that the authors should discuss including the role of PBN to POA neuronal populations in tail vasodilation. This function is assigned by Yang et al. to a population of PBN neurons that instead express cholecystokinin.

We thank the reviewers for this insight. We agree that there are some important considerations in that regard, and now address them here and in the revised manuscript.

In the work from Yang *et al.*, the authors report findings largely consistent with those we present, including the central finding that glutamatergic PBN neurons expressing Pdyn drive heat defense (Yang, Du et al., 2020). Yang *et al.* show that photostimulation of parabrachial Cck or *Pdyn* expressing PBN neurons drives a decrease in body core temperature. They observe increasing tail temperatures indicating of vasodilation in response to photostimulation of Cck but not Pdyn expressing PBN→POA terminals. Surprisingly, they do not report a change in tail temperature evoked by activation of PBN Pdyn+ neurons despite decreases in core body temperature greater than those seen by activation of Cck expressing terminals. Here, we report rapid rise in tail temperature in response to photostimulation of VGLUT2^+^, Pdyn+, and PenK^+^ PBN→POA terminals. In response to reviewers’ suggestion, we examined the overlap of *Pdyn* and *Cck* expression in the PBN using *in situ* hybridization. In *Pdyn* labeled cells, we found frequent co-labeling by *Cck* probes (Figure 2—figure supplement 2). Findings from these experiments indicate 70% of *Pdyn* labeled neurons are also labeled by *Cck* probes. This congruency helps to resolve any discrepancy that a majority of these two neuronal populations are indeed overlapping.

2) With respect to defining the PBN aspect of the circuit, the authors should report more precisely the scope of the c-Fos expression within the dynorphin, enkephalin and VGLUT2 populations. For example, they should express the c-Fos positive population as a fraction of Dyn- and Enk-positive PBN populations. This would reveal the possibility that some dynorphin neurons in this region may project to other areas and have other functions. The authors should mention/discuss the potential heterogeneity of this population. Related to this, are all of the EnK^+^ neurons that project to the POA Dyn+? A Venn diagram could help clarify the relationship of the four markers (Dyn, VGLUT2, c-Fos, Enk). Finally, the authors should include whether Dyn or Enk expression in PBN changes over development. Concerns relate to whether tomato expression differs from Dyn, Enk, or actual Cre expression in the adult.

This is an important point and clarification strengthens the manuscript’s findings. In agreement with the reviewers’ suggestion, the warm-activated (Fos positive) PBN neural population is a subset of the larger Pdyn expressing PBN neuron population. Our findings showing that subsets of PenK^+^ and Pdyn+ neurons are positive for Fos after warmth exposure are consistent with results reported by Geerling et al. (Geerling, et al., 2016) and findings indicating a diversity of roles for PBN Pdyn expressing neurons. Chiang *et al.* recent reported that Pdyn cells represent a quarter of the excitatory PBN neurons (Chiang et al., 2020). The presented sagittal image of Pdyn+ PBN neuronal projections (Figure 2—figure supplement 1K) shows likely projections of PBN Pdyn+ neurons to multiple brain regions including accumbens, regions of the thalamus, bed nucleus of the stria terminalis, and hypothalamic areas. Recent anatomic studies also found diverse projection targets for Pdyn expressing PBN neurons (Huang, Grady et al. 2020). Forthcoming work from the Bruchas lab implicates PBN neurons in modulating eating behaviors (Bhatti, Luskin et al., 2020).

We have expanded our discussion of functional diversity of PBN Pdyn expressing neurons including recent work indicating a role in aversive learning (Chiang, Nguyen et al., 2020) and in regulating feeding (Kim, Heo et al., 2020). To improve the overall clarity of paper we have added an additional figure (Figure 8) outlining the relationships of PBN neural populations and the behavioral and physiologic outputs of circuit activation. We think this makes it more clear how these various systems are integrated across the circuit, along with what we reported in this study.

To examine the question of whether all PenK^+^ PBN neurons projecting to the POA are positive for Dyn, we used *in situ* hybridization to examine the expression of *Pdyn* and *Penk* in POA projecting PBN neurons. We found that of cells in LPBN labeled by retroAAV-eGFP (POA projecting), 49 ±4% (mean ± SEM) were also labeled by *Penk* and *Pdyn* probes (Figure 2—figure supplement 2). Of the remaining GFP labeled LPBN neurons, 26 ±2% were labeled by either *Pdyn* or 12 ±1% (mean ± SEM) by *Penk*, but not both. 13 ±3% (mean ± SEM) of GFP labeled LPBN neurons were not labeled by either (Figure 2—figure supplement 2C).

With respect to the possible change in the expression of Pdyn in the PBN during development, this is a potential confound to defining the Pdyn expressing population in PBN using a permanent recombination marker line like the Ai14 line used in the present study to generate the results pertaining to warmth activation of PBN neural populations presented in Figure 1. This particular mouse line was previously well characterized using in situ by our group. We found in those experiments, that the line was largely protected from leaky A14 expression, and that Pdyn expression with Ai14 colabeling, aligns with wild-type in situ from our group and the Allen Institute(Al-Hasani, McCall et al., 2015). Finally, there is also limited evidence for dynorphin gene expression during critical development and cell fate decisions, which could create non-selective Ai14 reporting. Nevertheless, we are aware of this limitation and have added discussion to acknowledge the possibility.

The results here showing an overlap of Pdyn expression and activation by warmth are similar to earlier reports (Geerling, Kim et al., 2016) and results obtained by RNA sequencing in the work by Yang *et al.* (Yang, Du et al., 2020). The remainder of the studies presented here utilize viral injections into adult animals, and concerns regarding changing expression of Pdyn during development are not applicable.

3) With respect to defining the POA aspect of the circuit, the authors are focused on VMPO, but based on the data presented, it appears that other nearby regions may also have been targeted in their experiments (Figure 3C)? If so, what happens to body temperature when optogenetically activating those extra-VMPO Dyn-fiber terminals? MnPO/AVPe and OVLT have been implicated in osmo- and thirst regulation and may modulate heart rate (as described in cited papers and reviews), which could affect body temperature. Did the authors measure heart rate change upon optogenetic stimulation of PBN (Dyn/Enk) fibers terminating in the VMPO?

We appreciate this concern. To clarify further, the median preoptic nucleus (MnPO) , anteroventral periventricular nucleus (AVPe) and vascular organ of the lamina terminalis (OVLT) are located medial to the VMPO, and ventral portions on MNPO and portions of AVPe and OVLT were likely covered in the light cone from the midline placement of optical fibers as illustrated in Figure 4B. To further aid in clarity, we added outlines demarcating MnPO and OVLT to Figure 3.

Considering the additional physiological impact of these manipulations, increased heart rate occurs in response to cold as part of increased metabolism to support thermogenesis (Nakamura and Morrison, 2007). Heart rate may change in response to activation of the PBN→POA projections but whether this is a primary or secondary effect due to the hypothermia would require substantial carefully designed experiments targeted specifically at this interesting question. In the presented studies on heat defense, we did not include an examination of heart rate change in our studies on thermal defensive behaviors and body temperature regulation. However, the hypothesis that activation of Pdyn+^PBN→POA^ projections inhibits cold evoked increases in HR is an exciting avenue for future studies.

4) Data demonstrating the requirement for the circuit is weakened by an apparent lack of specificity in chemogenetically inhibiting the VGLUT2 population. Do VGLUT2^+^ neurons in this area project elsewhere? Specific inhibition of the Dyn+ neuron-POA circuit would be more appropriate.

This is an important point the reviewers make regarding specificity of the chemogenetic manipulations. Therefore, in response to reviewer comments, we undertook new experiments using heat challenge in Pdyn-Cre mice expressing inhibitory Gi DREADDs in the Pdyn+ PBN neurons to address this question. Activation of Gi DREADDs by CNO did not significantly alter tail vasodilation is response to thermal challenge (Figure 5—figure supplement 1). Gi DREADD mediated inhibition of Pdyn+ PBN neurons was not sufficient to block thermal challenge evoked tail vasodilation. We interpret these findings to suggest that, as represented in Figure 8, the entirety of the heat defensive neural population in PBN is VGLUT2^+^, whereas only a subset of this population is Pdyn+ (Figure 1F). The remaining warm-activated LPBN neurons not blocked by inhibition of the Pdyn+ population may be enough to drive heat defensive tail vasodilation. We have added more to the Discussion to address these new findings in a scholarly manner.

5) The authors frequently mention a role for opioids in the POA in thermoregulation and perform an experiment to test whether dynorphin and enkephalin in this particular PBN-POA circuit has a role. It may be the case that the peptides released by these projections do not have a role, but the authors need to demonstrate positive controls for the antagonists. Are they able to cross the blood brain barrier and was there sufficient time for the drugs to enter the brain before optogenetic stimulation? The authors could also consider trying other antagonists particularly for KOR such as DIPPA (Cintron-Colon et al., Current Biology, 2019).

In the presented studies we employed three separate opioid receptor antagonists with different selectivity for opioid receptors. All have been previously used in vivo, well studied and established to cross into the CNS. Our laboratory has substantial expertise in opioid neuropharmacology and have used and published with of all these ligands across a host of behavioral assays. We chose these ligands based on the consensus in the field. We note the following:

Naloxone is a non-selective opioid receptor antagonist with higher affinity for the μ-receptor and crosses(Goldstein and Naidu 1989, Dean, Bilsky et al., 2009). Naltrexone is a more potent opioid receptor antagonist and has a higher relative affinity for κ*-*receptors over μ-receptors. It crosses into the brain rapidly with high levels(Wang, Raehal et al., 2004). Norbinaltorphimine (norBNI) is a long-acting *k-*receptor selective antagonist that actively penetrates the CNS (Bruchas, Yang et al., 2007). To further validate that the doses and time courses we used were effective, we examined the block of suppression of locomotion by the *kappa* receptor agonist U50,488 by naltrexone and norBNI (Paris, Reilley et al., 2011).

To show a positive control for norBNI’s selective action at KORs under the conditions we used here, we did an experiment, whereby pretreatment with naltrexone or norBNI was effective in reducing U50 (selective KOR agonist) mediated suppression of locomotor activity (Figure 4—figure supplement 1D-F). The presented pharmacology data indicate that acute heat defense behaviors are not mediated by engagement of opioid signaling. Opioid signaling may be implicated in balancing system tone and matching thermal demands, metabolism, and calorie intake, and is an area of active ongoing investigation.

[Editors' note: further revisions were suggested prior to acceptance, as described below.]

The manuscript has been improved but there are some remaining issues that need to be addressed before acceptance, as outlined below:The reviewers appreciate the importance of this work to understanding the neural circuitry of thermal defense behaviors and some of the revisions that were made by the authors did help clarify the manuscript, as outlined in attached reviews. On the other hand, a couple of the major issues raised by the reviewers were not adequately addressed (#1 and #2 in previous review). 1) The authors have advanced rather than resolved a discrepancy between their new data and that of Yang et al. (Science Advances) regarding Cck and Pdyn expression, with the authors claiming considerable overlap and Yang et al. claiming distinct populations. It is important to clarify this point and also with respect to the physiological mechanisms of thermal defense behavior. As it stands now the authors have presented a contradiction. 2) The percentage of Pdyn+ neurons that expressed c-Fos with warming and the percentage of Pdyn+ neurons that project to POA that expressed c-Fos with warming should be reported. The same should be reported for PenK^+^ neurons. Results from this analysis could alter the representations shown in Figure 8. 3) It should be noted that the new data in Figure 2—figure supplement 2 in theory provide important new information, but the specific signals within the images are not robust making interpretations of them difficult. i.e. There is one GFP cell clearly visible in panel B and what is background for panel E?

1) In sum, the results we report are highly consistent with results from a parallel, now published, studies by Yang et al(Yang et al., 2020). This is particularly noteworthy considering the studies were conducted by independent groups working in parallel located in separate countries and the results are highly consistent including the magnitude of changes in body temperature. Differences in results are, however, present between the two studies. The key differences are most relevant to one of the conclusions drawn by Yang *el al.* that our findings do not support.

The first notable difference in results is regarding populations of neurons expressing Pdyn and Cck. Yang *et al.* use immunohistochemistry (IHC) to localize Cck and Pdyn protein expression in the PBN. Their reported data shows minimal cell by cell overlap of labeling for Pdyn and Cck. Neurons labeled for either peptide are present in the lateral PBN with more broad expression of Cck in the PBN. The use of IHC to identify peptide expression is well established to be technically challenging for numerous neuropeptides. As a laboratory focused on them, we’ve ultimately decided in the past 5 years to forgo using IHC and antibodies to classify peptide-containing neurons due to our inability to be consistently confident in what the method tells us. Selectivity of antibodies, expression levels, protein-peptide levels changing under various condition, etc..all of these reasons are reported confounds of the techniques required (Fritschy, 2008; Gautron, 2019; Nassel and Ekstrom, 1997). Yang *et al.*, used colchicine treatment to disrupt normal peptide trafficking, concentrating the peptide in the soma, and this is often required for IHC of peptides, such as dynorphin.

In contrast, we employed fluorescent in situ hybridization (FISH) to examine *Cck* and *Pdyn* transcript expression in PBN to avoid challenges with IHC, and for higher resolution of genetic markers of cell identity. Even in RNAseq sutides, in situ is ofen required as a post-hoc validation and more closely represents “ground truth” for identification of a particularly genetic cellular identity. One thing we also mention, is that even though we used in situ, our findings are largely consistent with those reported by Yang *et al.* with respect to distribution and anatomic localization. We did, however, identify that *Cck* is co-expressed in 70% of Pdyn expressing PBN neurons (Figure 2—figure supplement 2) A further factor relevant to addressing this specific discrepancy in results is potential changes in expression levels of the peptides. The amount of protein produced and stored in the cell for a given peptide may vary with time and conditions. Therefore, we think it is certainly reasonable to presume that a potential source of the difference in finding overlapping vs separate Cck and Pdyn PBN neural populations results may rest with biological variables or differing techniques used to examine Pdyn and Cck expression in each study. We have now noted these caveats in the Discussion.

A second point where the results in Yang *et al.* and our results differ is in vasodilation and suppression of BAT in response to activation/suppression of Pdyn+ or CcK^+^ PBN→POA neurons, respectively. They report that activation of Pdyn+ PBN neurons induces hypothermia (similar to our results) but does not lead to vasodilation when done with Gq DREADDs or activation of POA terminals with photostimulation. They reported photostimulation for examining tail vasodilation used is by Yang *et al.* is the 20Hz alternating on and off at 2 seconds intervals and they used a completely different opsin (ChIEF vs hChR2(H134R)) construct to mediate photoactivation. The differences in these opsins are important because they could drive release of different transmitters and thus impact down-stream effects in the POA with various outcomes. Yang *et al.* report that tail vasodilation in response to activation of CcK^+^ PBN neurons by DREADDs and photoactivation of PBN→POA terminals. They further report that blocking transmission of Pdyn+ neurons lead to BAT activation but blocking CcK^+^ neurons did not have the same effect. We found hypothermia, vasodilation, and suppression of BAT activity in response to activation of VGLUT2^+^, Pdyn+, or PenK^+^ terminals in response to 10hz stimulation of PBN→POA terminals.

We agree with the reviewers that implications of these differences are worth discussing. The Yang *el al*. study did not examine distinct roles for Cck or Dyn peptides in neuromodulation. The expression of the peptides served only as markers to narrow the involved PBN→POA neural populations. From their data they argue for separation of the neural circuits involved in heat defense by modulating BAT and vasomotor tone at the level of the PBN but not separate roles for the peptides themselves. We found no roles for Dyn or Enk signaling in mediating the acute effects of PBN→POA neural activation. Our data indicate instead, that neither *Pdyn* nor *Penk* expression mark a functional separation in heat defense neural circuitry at the level of the PBN. Thus, the data we report does not support one conclusion from Yang *et al.,* specifically, that Pdyn expression in PBN→POA neurons marks a functional (or categorical) separation in thermal defense circuitry. Our studies do not address if such separation may or may not exist beyond Pdyn+ and PenK^+^ PBN →POA neurons. Taken together, the two reports can be used together make predictions to be tested in future studies and further delineate thermal regulatory neural circuits. For example, identification of separate neuronal population in the POA that project differentially to distinct nuclei and received inputs from separable PBN populations would support the functionally distinct pathways that are separate at the level of the PBN. We also note that the authors of the Yang paper contacted us unsolicited and were scholarly and accepting of the potential differences and similarities in our studies, and results.

Regarding the conclusions we draw in our report, the data presented in Yang *et al.* are only discordant with respect two points. First, that Dyn and Cck expressing PBN populations are distinct. We see overlapping expression at the level of transcripts by FISH and they however indicate separation at the cellular level in protein expression using IHC. Second, we observe vasodilation of the tail in response to activation of Pdyn+ PBN→POA neurons and they do not. Because they do not observe tail vasodilation in response to activation of PBN→POA Pdyn+ neurons it is unclear how the persistence of warmth induced tail vasodilation despite toxin-based blockade of Pdyn+ POA neurons should be interpreted within this context but it does fit with our data. We found that the Gi DREADD based suppression of Pdyn+ PBN neurons did not block warmth induced tail vasodilation.

Our report and the paper from Yang *et al.* are remarkably consistent for parallel studies from completely independent groups. The differences as we note above, can be attributed to experimental details, different techniques, or underling biological variables. We have now added further consideration of these points to the discussion to ensure that the field is aware of these differences, and provide ideas for further exploration using high resolution approaches including RNAseq in PBN-POA projection neurons – a massive, albeit, fascinating potential future direction, which will be more definitive in regard to cell type segregation in this region. While we cannot completely resolve the focal difference in the data in our report and those in the Yang *et al.* manuscript, we did expand the discussion of this difference in the Discussion section. The differing results do lead to testable hypotheses that can be examined in future studies on thermal circuits.

2) The reviewers ask that “the percentage of Pdyn+ neurons that expressed c-Fos with warming and the percentage of Pdyn+ neurons that project to POA that expressed c-Fos with warming should be reported.” As we detail below, the experiments we can undertake to address this question would yield a quantification, but experimental factors present confounds that make a meaningful interpretation which would answer the underlying question highly unlikely.

The goal of quantifying the percentage Pdyn+ or PenK^+^ PBN cells and/or Pdyn+ or PenK^+^ PBN→POA neurons activated by warmth is highly unlikely to be meaningfully answered due to technical confounds of available methods and limitations inherent to interpretation of Fos staining as a marker of neuronal activity. If we understand the reviewers concerns correctly, they are asking, do all (or nearly all) Pdyn+ or PenK^+^ PBN→POA projecting neurons respond to environmental warmth? The option for attempting to address this question that we see as tenable would is injection of retroAAV-DIO-eFYP (or other marker) into the POA of Dyn-Cre and Penk-Cre mice to label Pdyn+ or PenK^+^ expressing PBN→POA neurons, exposing the animals to environmental warmth, and then probe PBN containing sections for Fos expression. Although this experiment would yield a relative quantification, meaningful interpretation of potential results would be confounded by multiple variables. For example, retrogradely labeled PBN neurons not positive for Fos staining may be considered not warmth responsive, but it would remain unknown if treatment with environmental warmth at a higher temperature or a more prolonged time would induce Fos expression in the cells. The time course and changes therein are not trivial to work out in a careful manner within the scope of this report. Further, cells positive for Fos labeling but not labeled by the retrograde virus could be interpreted as not projecting to the POA. However, such cells may simply have not been labeled by the retrograde virus but do in fact project to better POA. As an additional example, cells not directly involved in thermal heat defense could also be activated in response to environmental warmth due the effects of warmth on multiple physiological parameters. In the context of the PBN and POA, which drive a wide range of homeostatic and behavioral processes, this is particularly relevant. As these example illustrates, clear quantification to address what we understand to be the underlying question from the reviewer is likely to yield a number, but the meaning of the quantification would remain unclear. It is likely that projection-specific RNAseq and other high resolution physiological measures like in vivo imaging would be better suited for this experiment, yet as the reviewers might imagine, these are beyond the scope of this initial report.

The requested quantification of Fos positive cells from the Pdyn-Cre crossed to a recombination reporter line (Fos+/Pdyn+) has been performed previously in published articles (Geerling et al., 2016). Repeating that analysis here where add little total value and such experiments would be confounded by similar concerns as those identified above regarding Fos staining as a marker. As a further barrier, we no longer have the require mouse lines and the original slides are now many years old.

The reviews indicate that these results might alter the graphic illustration in Figure 8. We apologize for confusion derived from this cartoon. The illustration is meant as a Venn diagram to illustrate, in a logical framework, our findings of overlapping expression without making overt statements regarding quantification. In response to reviewers concerns we have edited the amount the circle representing the Pdyn+ neurons overlap with warm-activated cells to help avoid confusion about what may be implied.

3) With respect to questions about data shown in Figure 2—figure supplement 2, we appreciate the feedback regarding how the images are displayed. To address this we have now revised the figure by adding additional panels showing three areas of the PBN in greater detail.

We found these four-color FISH experiments, examining expression of two neuropeptides and retrogradely labeled neurons be technically challenging. The question regarding the single GFP positive cell, perhaps reflects concerns about overall displayed figure. The puncta from the FISH labeling for GFP can be seen in many of the cells and we have revised this figure to help make that more evident by displaying images at two levels of detail to convey anatomic and cellular level information. It is worth noting that we find the function of the fluorophore (GFP) is disrupted by the hybridization process and thus, the fluorescent signal is derived from the tags on the probes targeting GFP mRNA. We feel the additional panels demonstrate the background labeling was quite low in this experiment was low. This experiment was undertaken to address reviewer question about overlap of *Pdyn* and *Penk* expression in PBN→POA neurons.

Reviewer #1:The study by Norris et al. uses optogenetic /chemogenetic manipulations with physiological measurements and behavior as well as circuit tracing to demonstrate that excitatory neurons in the LPBN, some of which express Pdyn and Penk (partially overlapping), and that project to the POA are important for heat defense behaviors.1) Explain how their work fits with work by Yang et al., which is now published in Science Advances.2) Illustrate better the overlap of c-Fos within the three populations, whether all PenK^+^ neurons that project to POA are also Pdyn+ and whether Pdyn or Penk expression from the tomato mouse cross differs from adult in situ or adult virus injection (developmental change).

We agree with the reviewer’s points, and they make important comments with respect to our study as compared to the Yang report, that weren’t addressed as clearly in the original resubmission.

1) With respect to the overall findings in Yang *et al.* report compared to the data that we present; the results are very similar. Reviewer one has summed up nicely much of how the results are different. As we discussed above in our reply to summary comments, there are technical and experimental details which may account for some portion of the difference in findings; however, the results we obtained with respect to differential modulation of BAT activity and vasomotor tone by Pdyn+ neurons do not match those reported by Yang *et al.* Future studies designed to further delineate potential separation of neural circuits regulating aspects of thermal heat defense behaviors at the level of the PBN and POA may help resolve this narrow difference in the conclusions drawn by the two studies. Our study does not support a conclusion drawn in Yang *et al.,* that expression of Pdyn is a marker of this functional division in the neural circuitry at the level of the PBN. We have greatly expanded our Discussion of these differences and possibly implications in the revised manuscript.

2) In previous studies expression of tdTomato in Ai14 x Pdyn-Cre mice has been carried out in in the nucleus accumbens are high level of concordance was found between expression Pdyn and tdTomato (Al-Hasani et al., 2015). We have not had the cross of Ai14 to Pdyn or Penk mice for several years now and cannot perform the requested *in situ* hybridization in the parabrachial nucleus without a substantial delay required to breed and allow the required animals to come to adulthood. Review of the Allen Brain Developing Brain Atlas shows expression of *Pdyn* in the external PBN at P28. Experiments examining overlap of tdTomato expression with *Pdyn* transcripts in the PBN would one time of information about the stability through development we do not feel that results may be obtained would substantially alter interpretations of the results presented. Only data in figure one, which is largely consistent with Geerling et al., 2016, is impacted by this concern (Geerling et al., 2016). We feel we offer a conservative interpretation of this data. The remainder of the experiments are not impacted by concerns about changes in Pdyn during development.

Reviewer #2:The authors have not adequately addressed the reviewers' comments and concerns. This is not to say that we, as reviewers, expect the authors to do all the experiments --or for that matter even change all the wording in the text-- as per reviewers' suggestion. But authors should at least address in the point-by-point rebuttal letter and state why they disagree with the suggestions and/or (i) did not need to or (ii) couldn't or (iii) did not want to implement the changes the reviewers suggested.1) The authors suggest that Pdyn neurons in the PBN specifically relay temperature information to the POA/VMPO. If Pdyn is labeling neurons that are warm-activated, I would expect a substantial fraction of cfos-positive cells to overlap with pdyn. To assess this requires to calculate the cFos-positive population as a fraction of Dyn- and Enk-positive populations (e.g. what % of Dyn neurons is cfos positive upon warming) instead of expressing the cfos-fraction that is positive for dyn or enk (as has been done by the authors).In the response the authors vaguely argue that this information can be found in Geerling at el 2016. We looked it up and Geerling et al. suggest that around 20% of pdyn neurons are specifically induced to express cfos upon warming. Comparing the figure in Geerling et al., 2016 (Figure 7) with the equivalent one in this manuscript (Figure 1D) my best guess is that in the hands of the authors this fraction is even lower. And thus this fraction is misrepresented in the summary Figure 8: here in the cartoon it looks like the majority of warm activated cells are pdyn positive! This also renders the following sentence in the Discussion questionable "...A subset (Pdyn+ or PenK^+^) of the VGLUT2^+^ PBN→POA population is likely sufficient to mediate vasodilation and suppress BAT activation.." We disagree, in our view this means that the largest fraction of pdyn neurons is likely doing something else and not relaying temperature information. This possibly explains also the new data in Figure 5—figure supplement 1: blocking the pdyn population by Gi-DREADD does not have an effect (different to blocking vglut2-Cells).Reversely, does this mean that their Gq-DREADD data (Figure 4I) is wrong and activating the pdyn population should also not activate tail vasodilation as suggested by Yang et al., biRxv or Science Advances 2020? Again, I like to reiterate that the authors don't need to do all what the reviewers suggest, but glossing over important points and not addressing them properly in the letter (and the manuscript) is not a good practice. This brings us to the next point:

We understand that two points of concern are raised in first comment. One addresses the presentation in the cartoon in Figure 8 of the potential for implied percentage of Pdyn+ POA neurons that are positive for Fos expression following warmth exposure(warm-activated). We agree with the findings from Geerling *et al.* that a minority of total Pdyn+ PBN neurons are positive for Fos following warmth exposure. Further, analysis of these experiments would, we feel, produce data which that be difficult to interpret for reasons detailed above. We have revised Figure 8 to reduce potential for confusion about the extent of the overlap of Pdyn and Fos staining after warm exposure.

With respect to the second point, we are less clear on what the reviewer is pointing out but would like to review a few of the points with the hope clarify. We find that photoactivation Pdyn+ and PenK^+^ PBN→POA terminals in the POA leads to tail vasodilation, hypothermia, BAT suppression, and aversion. We showed by FISH that neurons labeled by *Penk* and *Pdyn* probes also express *Slc17a6* (VGLUT2). From these findings taken together, we conclude that activation of PenK^+^ or Pdyn+ PBN→POA neurons, which are subpopulations of the VGLUT2^+^ PBN neuron population, is sufficient to drive tail vasodilation, BAT suppression, hypothermia, and aversion. Our studies focused on the PBN→POA neurons. In all activation experiments we applied light via fiber optic implanted over the VMPO. This was done to avoid photoactivation of PBN neurons that do not project to the POA. The data and conclusions we present make no claims to the functional roles of all or most PBN Pdyn+ neurons. None of the presented experiments were done with Gq-DREADDs as the reviewer suggests. The data in Figure 4I was obtained from experiments using 10Hz photostimulation in the POA of Pdyn+ POA→POA neurons.

We found that Gi-DREADD mediated inhibition of VGLUT2^+^ but Pdyn+ PBN neurons blocked tail vasodilation is response to thermal challenge. We interpret this finding to suggest that the Pdyn+ PBN→POA projecting population is a subset of the VGLUT+ PBN→POA population. As a technical point, this could be also due to a limited effect of the Gi signaling mediated inhibition.

2) The authors now do dual-color in situs to test what fraction of CCK neurons overlap with pdyn neurons to conclude that "..Findings from these experiments indicate 70% of Pdyn labeled neurons are also labeled by Cck probes. This congruency helps to resolve any discrepancy that a majority of these two neuronal populations are indeed overlapping…". We beg to differ: the authors Yang et al., 2020 use --as far as we can see-- the same pdyn-Cre mouse line as the authors and they don't see any tail vasodilation when activating these neurons (Figure 5B in the Science Advances publication) and this has nothing to do with any CCK expression. Again, this discrepancy may not be easily resolvable, but to hide it and to say CCK is a subset of these neurons and this resolves the discrepancy is not correct! This functional difference may --at least for some researchers- be important and thus should be spelled out.

The differences between the results we show and those from Yang et al. focus on the effect of activation of Pdyn PBN→POA neurons on tail vasodilation and if Pdyn and Cck expression are markers for functional division in the heat defense circuitry at the level of the PBN. We find that activation of Pdyn+ PBN→POA terminals results in tail vasodilation and suppression of BAT thermogenesis and leads to rapid hypothermia. Yang *et al.* do not report tail vasodilation in response to activation of Pdyn+ PBN→POA but do observe similar rapid onset in hypothermia despite the lack of active heat shedding via vasodilation. They do see tail vasodilation in response to activation of Cck PBN neurons and report that Cck and Pdyn expressing neurons are sperate populations based on IHC. Considering the technical challenges with IHC for Pdyn, we used FISH to examine expression of *Pdyn* and *Cck* in the PBN and found these two populations had substantial overlap. Thus, our data does not indicate there is a functional separation in heat defense circuity at the level PBN demarcated by Pdyn expression. Despite this narrow difference in the results overall, both reports are quite consistent. We have expanded the Discussion in the manuscript of these the divergent results in the manuscript and addressed the points further in the reply to the summary comments.

3) The manuscript is very difficult to read with all the edits, "invalid citations" etc.The authors should read the manuscript carefully and correct before submission.

Thank you for highlighting these concerns. We worked to correct and refine the manuscript and have minimized the edits evident in the uploaded version.

[Editors' note: further revisions were suggested prior to acceptance, as described below.]

The manuscript has been improved but there are some remaining issues that need to be addressed before acceptance, as outlined below:1) The comparison of the Pdyn staining and the Pdyn-Cre line shown in Yang et al. Figure 2A indicates a high level of co-localization. Norris et al. have used the same Pdyn-Cre line in the present manuscript. Comparison of Pdyn and Cck in Yang et al. Figure 2L was done by staining the Cck-Cre mouse with the Pdyn antibody used in Figure 2A. The difference between Yang et al. and Norris et al. is therefore the difference between the Cck-Cre line (Yang et al) and the use of RNAScope (current manuscript). The authors should correct how they discuss this in the manuscript, as it now is stated as being due strictly to antibody staining by Yang et al. vs RNAScope used by the authors of the current manuscript.

We have edited the text of the manuscript to further clarify the differences between how Yang et al. and we assayed for expression of Cck and Pdyn. As highlighted by the comments, we used RNAScope and Yang et al. used a combination if IHC (for Dyn) and viral Cre dependent reporter in the CCk-Cre mice.

2) As for showing the percentage of Pdyn and Penk neurons that express c-Fos based on data shown in Figure 1, an additional limitation (on top of those noted by the authors) to the interpretation of reporting cell counts as a percentage of either c-Fos or the marker would also be if the number of tomato+ cells observed using the Cre lines crossed to the reporter is not the same as the Pdyn or Penk population manipulated by the viruses. The authors addressed this concern in an earlier version in their response to the reviewers, but direct evidence in the paper is lacking. Nevertheless, providing the readers with some sense of the percentage of Pdyn or Penk neurons that were c-Fos positive from the images that they quantified in Figure 1 would be similarly useful.

We blindly sampled tdTomato expressing cells in the LPBN in brain sections from Ai14xPdyn-Cre or Ai14xPenk-Cre mice and then quantified the number of those cells that were also labeled for Fos induction after warm exposure. For Pydn+ neurons we found 22% of the tdTomato cells were also Fos+, similar to the finding of 27% by Geerling et al., 2016 in similar experiments. For PenK^+^ we found 18% of tdTomato cells were labeled for Fos. These values are now reported in the text.